# Cryo-electron microscopy of IgM-VAR2CSA complex reveals IgM inhibits binding of *Plasmodium falciparum* to Chondroitin Sulfate A

Reetesh Raj Akhouri [1,3] ✉, Suchi Goel[2] & Ulf Skoglund[1]

Placental malaria is caused by *Plasmodium falciparum*-infected erythrocytes (IEs) adhering to chondroitin sulfate proteoglycans in placenta via VAR2CSA-type PfEMP1. Human pentameric immunoglobulin M (IgM) binds to several types of PfEMP1, including VAR2CSA via its Fc domain. Here, a 3.6 Å cryo-electron microscopy map of the IgM-VAR2CSA complex reveals that two molecules of VAR2CSA bind to the $C\mu4$ of IgM through their DBL3X and DBL5ε domains. The clockwise and anti-clockwise rotation of the two VAR2CSA molecules on opposite faces of IgM juxtaposes C-termini of both VAR2CSA near the J chain, where IgM creates a wall between both VAR2CSA molecules and hinders its interaction with its receptor. To support this, we show when VAR2CSA is bound to IgM, its staining on IEs as well as binding of IEs to chondroitin sulfate A in vitro is severely compromised.

*Plasmodium falciparum* is a major malaria parasite that infects around 247 million humans and causes around 625,000 deaths every year in endemic areas, mostly children under the age of five[1]. While malaria is mostly associated with symptoms like chills and fever, *P. falciparum* infection also causes multi-organ pathogenesis because of its ability to sequester in various tissues and organs of infected individuals. Severe malaria pathogenesis is caused when IEs cytoadhere in the brain and placenta or form rosettes (auto-agglutinates of IEs and uninfected erythrocytes) that cause blockage of blood flow to tissues and organs. Sequestration in the brain leads to cerebral malaria (CM), and when IEs sequester in the placenta of pregnant women, it causes placental malaria (PM)[2,3]. Usually, the outcome of CM is more severe than PM; however, PM is still associated with complications to both mother and fetus, causing poor fetal growth, sudden abortions, low birth weight, and even death of both newborn and mother due to complications in pregnancy or anemia[4,5]. Therefore, understanding and unraveling the mechanism of PM pathogenesis is an urgent issue.

The *P. falciparum* protein PfEMP1(*P. falciparum* Erythrocyte Membrane Protein 1), expressed on the surface of IEs in a mutually exclusive manner, is responsible for the sequestration of IEs in various organs[6–8]. It has been well established that in the case of PM, the PfEMP1 member VAR2CSA binds to low sulfated chondroitin sulfate A (CSA) or chondroitin sulfate proteoglycan (CSPG) expressed on placental intervillous tissue to mediate sequestration of IEs in the placenta[9–11].

VAR2CSA is a >310 kDa protein comprising of multiple Duffy binding-like (DBL) domains and interdomains (ID) and shows specific and a higher binding affinity to CSA when compared with individual DBL and ID domains. Recent, cryo-EM structure of apo-VAR2CSA and bound to dodecamer CSA revealed that VAR2CSA mainly interacts with its receptor CSA via a binding site comprising a groove formed by DBL2X[12–14]. VAR2CSA also binds to the Fc domains of pentameric immunoglobulin M (IgM)[15] that is shown to inhibit the binding of DBL3X and DBL5ε-specific monoclonals IgGs, suggesting that the interaction of these domains with IgM could play a role in immune evasion. Human pentameric IgM in a low resolution cryo-AFM study revealed mushroom-shaped architecture that consists of 5 copies of monomeric IgM and a J chain (JC)[16]. Each monomer has two heavy and two light chains. Further, each heavy chain has a variable domain and Fc region (constant region) called $C\mu$ domains ($C\mu1$-$C\mu4$) and a tailpiece

[1]Okinawa Institute of Science and Technology Graduate University, Okinawa, Japan. [2]Indian Institute of Science Education and Research Tirupati, Tirupati, India. [3]Present address: Indian Institute of Technology Madras, Chennai, India. ✉e-mail: akhourirr@gmail.com

that are C-terminal extensions following the Ig-fold of Cμ4 (557–576 amino acid)[17–19]. Because IgM has multiple domains, it is important to ascertain which domains directly mediate binding to IgM and whether the interaction of the IgM affects the binding of VAR2CSA to its receptor CSA.

Here, we show that VAR2CSA expressing *P. falciparum* strain, CS2 binding to CSA is reduced in the presence of non-heat inactivated plasma (NHI-plasma- +IgM). In order to understand this observation, we used cryo-EM to derive a 3.6 Å map of an IgM-VAR2CSA complex, which reveals that IgM is sandwiched between two molecules of VAR2CSA. Our structure also shows that DBL3X and DBL5ε directly interact with the Cμ4 of IgM, possibly leading to the steric hinderance in VAR2CSA interaction with its receptor. Further, the addition of IgM to CS2 culture grown in albumax reduces the detection of VAR2CSA on IE surface as well as show a drastic decrease in VAR2CSA binding to its receptor in a manner similar to the NHI-plasma, suggesting that IgM inhibits VAR2CSA ability to bind to its receptor.

## Results

### VAR2CSA-expressing IEs binding with CSA is compromised in NHI-plasma

In order to understand the role of IgM in modulating the binding of VAR2CSA to CSA, we used *P. falciparum* strain CS2, which is known to maintain stable expression of VAR2CSA on IEs surfaces[20], and growth media containing albumax as it is devoid of IgM. After confirming the expression of VAR2CSA on the surface of CS2-IEs (Supplementary Fig. 1), we propagated CS2 in media containing either NHI-plasma (+IgM) or albumax (-IgM) and performed CSA binding assays. We observed that propagation of CS2 in NHI-plasma reduced binding to CSA by 50% as compared with CS2 cultured in albumax (Fig. 1a, b). We then tested whether this reduction in binding is due to lower expression of VAR2CSA in CS2 cultured in NHI-plasma by performing western blotting using anti-VAR2CSA antibodies (anti-PfHSP70 antibodies as a loading control). We observed that expression of VAR2CSA is equal in both conditions (Supplementary Fig. 1b), suggesting that the observed reduction in CSA binding in NHI-plasma is not due to lower expression of VAR2CSA, but due to IgM that inhibits the binding of IEs to its receptor.

### Cryo-EM structure of the IgM-VAR2CSA complex

Previously it has been shown that DBL2X, DBL5ε, and DBL6ε bind to IgM[21,22]. To further understand IgM interaction with VAR2CSA, we used cryo-EM to resolve the assembly of the IgM-VAR2CSA complex. We overexpressed and purified recombinant VAR2CSA to homogeneity using immobilized metal affinity chromatography and size exclusion chromatography (SEC, Fig. 1c). Further, we prepared, screened and characterized the IgM-VAR2CSA complex by performing 10–40% sucrose gradient fixation (GraFix) in the presence of 0.2% glutaraldehyde (Fig. 1c, d). To check for the formation of aggregates due to the crosslinker, we also performed a 10–40% sucrose gradient

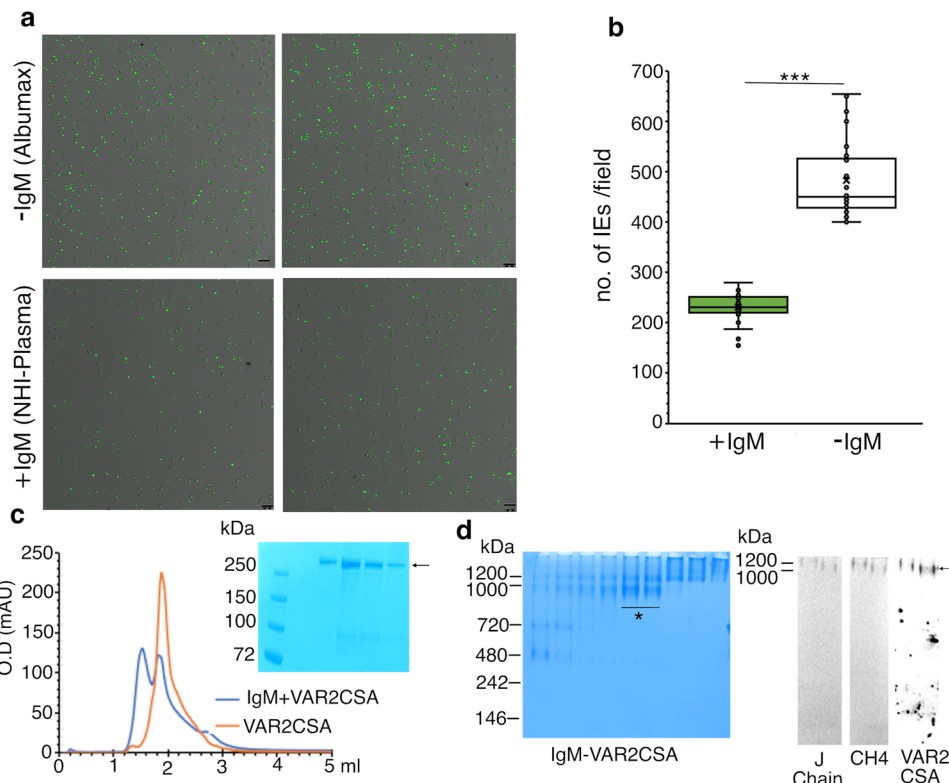

**Fig. 1 | Presence of NHI-plasma compromises cytoadherence of CS2 parasites to CSA. a** Representative images from three experiments of CSA binding of CS2 IEs cultured in RPMI either supplemented with NHI-plasma (+IgM) or albumax (-IgM). The bound IEs were stained with acridine orange and visualized under confocal microscope at 20X under bright field and alexa 488 filter and the merged images of +IgM and -IgM are shown. Scale bar is 25 μm. **b** The box-whisker analysis of bound IEs from merged images in +IgM and -IgM and plotted as number of IEs bound/field. $n = 3$ performed in duplicates and 5 fields were counted per spot and represented as a point in the box-whisker plot, cross (x) represents the mean, line crossing the box plot is the median, lowest and highest whisker represents minimum and maximum data point, whereas lower bound of the box represents Quartile 1 and upper bound of the box represents Quartile 3, dot outside the whisker are outliers (original values are available in source data). P values were calculated using paired t-test and ***P< 0.0001. **c** A representative image from five experiments of Size exclusion chromatography of VAR2CSA and IgM-VAR2CSA complex resolved on superose6 3.2/300 and the chromatograms were superposed. Orange-VAR2CSA peak, blue-IgM-VAR2CSA complex. The fractions containing VAR2CSA were resolved on SDS-PAGE and coomassie stained **d** A representative image from three experiments of the IgM-VAR2CSA complex fractions obtained upon gradient fixation resolved on native-PAGE. * represents the fractions used for western blot using anti-VAR2CSA, anti-CH4 and J chain antibodies. These fractions were also used to prepare cryo-EM grids for data collection on 300kEV Titan Krios.

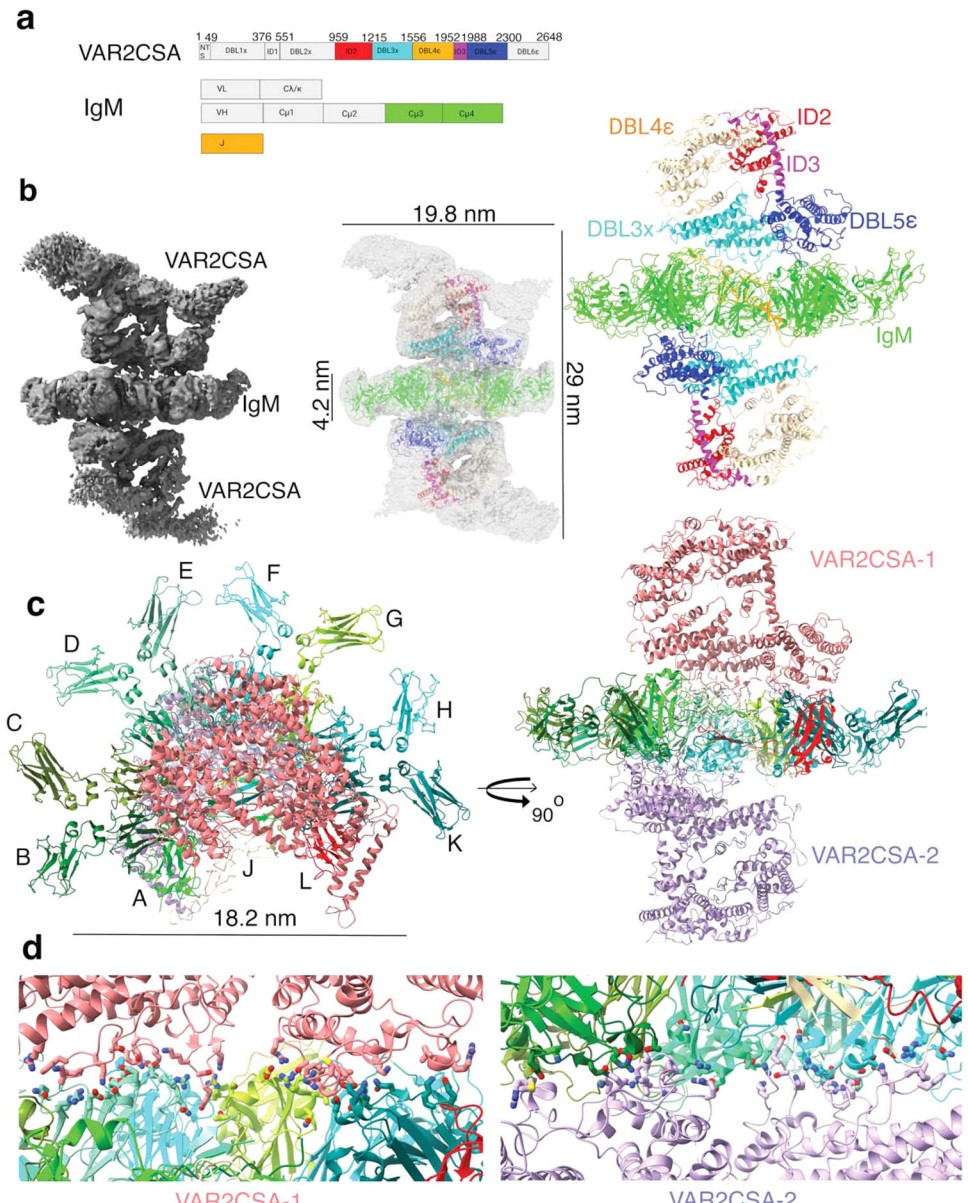

**Fig. 2 | Cryo-EM map of IgM-VAR2CSA complex at a 3.6 Å resolution.**
**a** Schematic representation of VAR2CSA and IgM domains **b** Cryo-EM map of IgM-VAR2CSA post 3D refinement (left). The map of the complex shows two VAR2CSA molecules bound to one IgM. The model of IgM-VAR2CSA complex fitted within transparent isosurface along with corresponding dimensions (middle). The resolved map of complex was modeled for domains of VAR2CSA(right), ID2(red), DBL3x (cyan), DBL4ε (brown), ID3 (magenta), DBL5ε (blue) and IgM is depicted as green. DBL1x, ID1, DBL2x and DBL6ε were not modeled. **c** The resolved map of complex modeled for IgM polypeptides as A (lime), B (green), C (olive drab), D(spring green), E(aquamarine), F (Cyan), G (green yellow), H (dark turquoise) and K (Teal), L (red) and J (brown) with VAR2CSA-1(Salmon) and VAR2CSA-2 (purple) as shown in orthogonal views. **d** The magnified view of the interfacing region consisting of DBL3x and DBL5ε of VAR2CSA-1 and VAR2CSA −2 along with the chains of IgM.

separation in the absence of the crosslinker (Supplementary Fig. 2). Upon ultracentrifugation and fractionation, we further resolved fractions on native-polyacrylamide gel electrophoresis (Native-PAGE) and observed similar patterns of the fractions irrespective of the crosslinker, indicating that the crosslinker did not lead to aggregation of protein complex (Supplementary Fig. 2). In order to further confirm the presence of IgM and VAR2CSA in the complex, we analyzed fractions of the IgM-VAR2CSA complex devoid of crosslinker on reducing sodium dodecyl sulfate PAGE and observed the presence of both VAR2CSA and IgM (heavy and light chains, Supplementary Fig. 2).

Our initial cryo-EM analysis of the fractions of the IgM-VAR2CSA complex obtained from GraFix suggested a homogenous distribution of complex in fractions 3 and 4. Therefore, after confirming the presence of both components in the fractions by western blot using anti-J

chain, anti-CH4, and anti-VAR2CSA antibodies (Fig. 1d), we prepared grids and collected cryo-EM data of the cross-linked IgM-VAR2CSA complex and derived a 3.6 Å map. The map consists of one IgM sandwiched between two VAR2CSA molecules (Fig. 2a, b, Supplementary Fig. 3, Supplementary Table 1). The overall dimensions of the resolved map is 29 nm (longer axis) × ~20 nm (shorter axis) × 18 nm (depth). While the respective dimensions of IgM are 18.2 nm × 4.2 nm × 18 nm, the dimension of VAR2CSA in the complex along the planar axis of IgM is 19 nm (Fig. 2b, c). To produce an atomic model, we used apo-VAR2CSA (Protein Data Bank [PDB] #7B52) to fit VAR2CSA in the cryo-EM map of the complex, which revealed the structures of the DBL and ID domains of VAR2CSA (Fig. 2b). Since there are two VAR2CSA molecules, we assigned chain I and M to the two VAR2CSA polypeptides (Fig. 2b).

The pentameric IgM that we used in the formation of the IgM-VAR2CSA complex is an approximately 1MDa molecule. We used PDB #6KXS to build an atomic model of IgM in the resolved map and assigned chains IDs A, B, C, D, E, F, G, H, K, and L to each heavy chain Cμ3-Cμ4 peptides of IgM and J to the J chain[23] (Fig. 2c). It is worth noticing that although we used whole IgM, yet out of all these domains of IgM, we could resolve only its planar central part of IgM to a high resolution, which consists of only part of the heavy chain of IgM (mostly Cμ3-Cμ4) and tailpiece (Fig. 2c) probably due to high flexibility as reported in domains beyond this region of IgM[24].

Our high resolution data at the center of IgM revealed that the tailpiece residues 562–568 that are rich with β-sheets from all poly-peptides assemble at the center of IgM, which is important for oligo-merization of IgM (Supplementary Fig. 4a). β-sheets in the tailpieces of peptides A−E are arranged parallel to each other. Similarly, β-sheets in the tailpieces of peptides F−L are also arranged in parallel (Supplementary Fig. 4b). However, one parallel set of β-sheets from the A−E stack antiparallel to the other set of F-L stack (Supplementary Fig. 4b). This stack of 10 β-sheets is essential for oligomerization and the sta-bility of IgM which also leads to the formation of the core of IgM. Additionally, this arrangement brings cys414 in Cμ3 from adjacent heavy chains to face each other, permitting the formation of a disulfide bond that further stabilizes the Cμ3-Cμ4 and tailpiece into a near planar structure[24,25]. This unique organization and limited degree of flexibility in the core of IgM allowed for high-resolution structure of these domains. This oligomerization is also important for the formation of two identical surfaces on the opposite faces of IgM (Supplementary Fig. 4a and b) to which two molecules of VAR2CSA bind through the Cμ4 domain.

**IgM interacts with VAR2CSA through domains DBL3X and DBL5ε**
Since 2 VAR2CSA interact with one IgM, we generated two pdbs by splitting the IgM-VAR2CSA pdb, where both contained IgM but in the opposite orientation and along with that they contained either chain I or chain M and called as IgMVAR2CSA-1 (lacking chain M) and IgMVAR2CSA-2 (lacking chain I). Then we compared these two pdbs using chimeraX matchmaker with the iteration cut off of 2 Å for pruning residues. Since the IgM is in opposite orientation in the pdbs used, we compared chain A of IgMVAR2CSA-1 to L of IgMVAR2CSA−2. Similarly, we compared their corresponding chains of IgMVAR2CSA-1 with IgMVAR2CSA-2 respectively; chain B to K, chain C to H, chain D to G, chain E to F, chain F to E, chain G to D, chain H to C, chain K to B, chain L to A and chain M to I of VAR2CSA. The RMSD between 2696 pruned residues out of total 3261 is 0.995 Å. However, for all 3261 residues, the total RMSD is 6.98 Å, mostly due to the J chain (Supple-mentary Fig. 5). RMSD across most of the interacting residues and VAR2CSA is very low (Supplementary Fig. 5). Further, when we ana-lyzed buried surface area using PDBePISA both VAR2CSA and IgM junction have a total buried surface area of ~1350 Å² on each face, further suggesting similar interfaces[26]. We build the model of the complex and observed that the interaction was mostly in the loop regions of the DBL3x and DBL5ε domains of VAR2CSA (Fig. 2b, d). We then wanted to test if the binding of IgM induces conformational changes in the DBL domains of VAR2CSA. For that we first stitched pdb 7B52 and pdb 7NNH in the map (chain X) in apo-VAR2CSA map[12,13]. We then compared apo-VAR2CSA (chain X)[12,13] with IgM-VAR2CSA (with only chain M) in the complex. We observed that although they have an overall RMSD of 1.367 Å, both structures align well between the ID2 to ID3 domain but have major changes and movement of corresponding residues in the DBL5ε domain (Fig. 3a, Supplementary Fig 6a). We calculated the distances of corresponding amino acids in X and M chains. On comparison with apo-VAR2CSA, we observed that Arg²⁰⁵⁵ that interacts with IgM and Tyr²⁰³³ that is proximal to the interacting surface shifted by a distance of 18.9 Å and 43 Å, respectively that placed these residues closer to IgM. While, amino acids Ser²⁰⁶² and

Asn²¹³⁶ moved by 23.7 Å and 42 Å, respectively, towards ID3 helix. Similarly, Leu²⁰²¹ moved by 32 Å, Asn²⁰²⁵ by 35 Å, Glu²⁰⁸⁷ by 20 Å, Lys²¹¹⁰ by 26 Å and Thr²¹⁸⁰ by 19 Å towards ID3 (Fig. 3a and Supplementary Fig. 6b). However, Ala²²⁸⁰ that is distant from the interacting surface showed a large movement of 55.8 Å away from ID3 (Fig. 3a), reflecting the huge shifts in the C-terminal domains of VAR2CSA and also sug-gesting that the regions away from IgM shift more than the region that are closer to the interfacing surface (Fig. 3a, b, Supplementary Fig. 6b). This results in the formation of a closed architecture in C-terminus of VAR2CSA in the IgMVAR2CSA map that differs from the apo-VAR2CSA architecture. Although, we did not model DBL6ε domain (Fig. 2b) due to the lack of resolution in this region of the map but the map for DBL6ε in the IgM-VAR2CSA complex seems to interact with the DBL4ε-ID3 junction, leading to the formation of a closed core comprising ID3-DBL5ε-DBL6ε (Fig. 2b). In the complex, ID3 separates ID2-DBL3X-DBL4ε core from DBL5ε-DBL6ε. ID2, which is connected to DBL3X through a linker, is also a highly important component of VAR2CSA as it links DBL2X and DBL3X and is also placed central to DBL4ε, DBL3X, and ID3. DBL3X is spread out and is proximal to IgM, making maximum contact with IgM. It is clear from the model that both DBL3X and DBL5ε are co-planar with respect to IgM core, unlike apo-VAR2CSA, primarily due to the realignment of C-terminus domains of VAR2CSA (Fig. 3a, Supplementary Fig. 6b).

**Three important interfaces stabilize IgM and VAR2CSA interaction**
We used PDBePISA[26] to analyze interfacing and interacting residues in the IgM-VAR2CSA complex. Further from the overall view of the interacting surface (Supplementary Fig. 7a), we observed inter-action between chains E, G and K of IgM with I chain of VAR2CSA (Fig. 3b) and chains F, D and B chain of IgM with M chain of VAR2CSA (Supplementary Fig. 7b). Also, chain E of IgM interacts with DBL3x, chain G of IgM interacts with both DBL3x and DBL5ε and chain K of IgM interacts with DBL5ε (Fig. 3b). Similarly, chain F of IgM interacts with DBL3x, chain D of IgM with DBL3x and DBL5ε and B of IgM with DBL5ε, suggesting very similar interactions on both faces (Supplementary Fig. 7b).

For interacting residues, we show that a total of 514 Å² area interfaces between chain I of VAR2CSA and chain E of IgM. Here, Arg⁴⁹¹ of chain E forms hydrogen bond with Pro¹²⁴¹ of chain I, Glu⁴⁶⁸ of chain E with Gln¹²³¹ of chain I, Thr⁵³⁰ and Glu⁵³² of chain E with Lys¹²³⁸ of chain I. Further, Glu⁴⁶⁸ of chain E forms an interface with Arg¹²²⁸ of chain I, Asn⁴⁶⁵ of chain E with Tyr¹²⁸² of chain I, Gln⁴⁹⁰ and Gln⁴⁹³ of chain E with Pro¹²⁴¹ of chain I, Glu⁴⁶⁸ of chain E with Gln¹²³¹ of chain I (Fig. 3b). Similar interaction between these residues are observed between DBL3X of chain M and chain F of IgM with the interface surface area of 468 Å² on the opposite face of IgM (Supplementary Fig. 7b).

Second junction is formed between chain G of IgM and chain I of VAR2CSA with a total of 418 Å² of interface area. Here, both DBL3X and DBL5ε contribute towards the junction formation with IgM. Here, Arg⁴⁶⁷ of chain G forms hydrogen bond with Asn²⁰⁵⁹ of chain I, Glu⁴⁶⁸ of chain G with Ala²⁰⁵⁸ of chain I. Additionally, Glu⁵²⁶ of chain G interacts with Arg²⁰⁶¹ of chain I, Leu⁴⁶⁶ of chain G with Pro²⁰⁵⁷ of chain I, Arg⁴⁹¹ of chain G with Ser¹²⁸¹ and Tyr¹²⁸² of chain I, Gln⁴⁹³ of chain G with Arg¹²⁸⁵ of chain I (Fig. 3b). We also observed similar interaction between chain D of IgM and chain M of VAR2CSA with a 384 Å² interfacing area between them (Supplementary Fig. 7b).

Chain K of IgM forms interface with the DBL5ε of chain I VAR2CSA. Here, Gln⁴⁹⁰ and Gln⁴⁹³ of chain K interacts with Lys²¹²¹ of chain I, Glu⁴⁹⁸ of chain K with Arg²⁰³⁰ of chain I, Glu⁵²⁶ and Thr⁵³⁰ of chain K with Arg²⁰⁵² of chain I, Glu⁵³² of chain K with Arg²⁰⁵⁵ of chain I. Additionally, Arg⁴⁹¹ of chain K is buried in a hydrophobic pocket surrounded by Ile²¹¹⁸, Ile²⁰⁵³, Val²⁰⁵⁴ and Gly²⁰⁵⁶ of chain I (Fig. 3b). These interaction together make a total of 300 Å² interfacing area. Again we observe a similar interaction on the opposite face of IgM between chain B of IgM

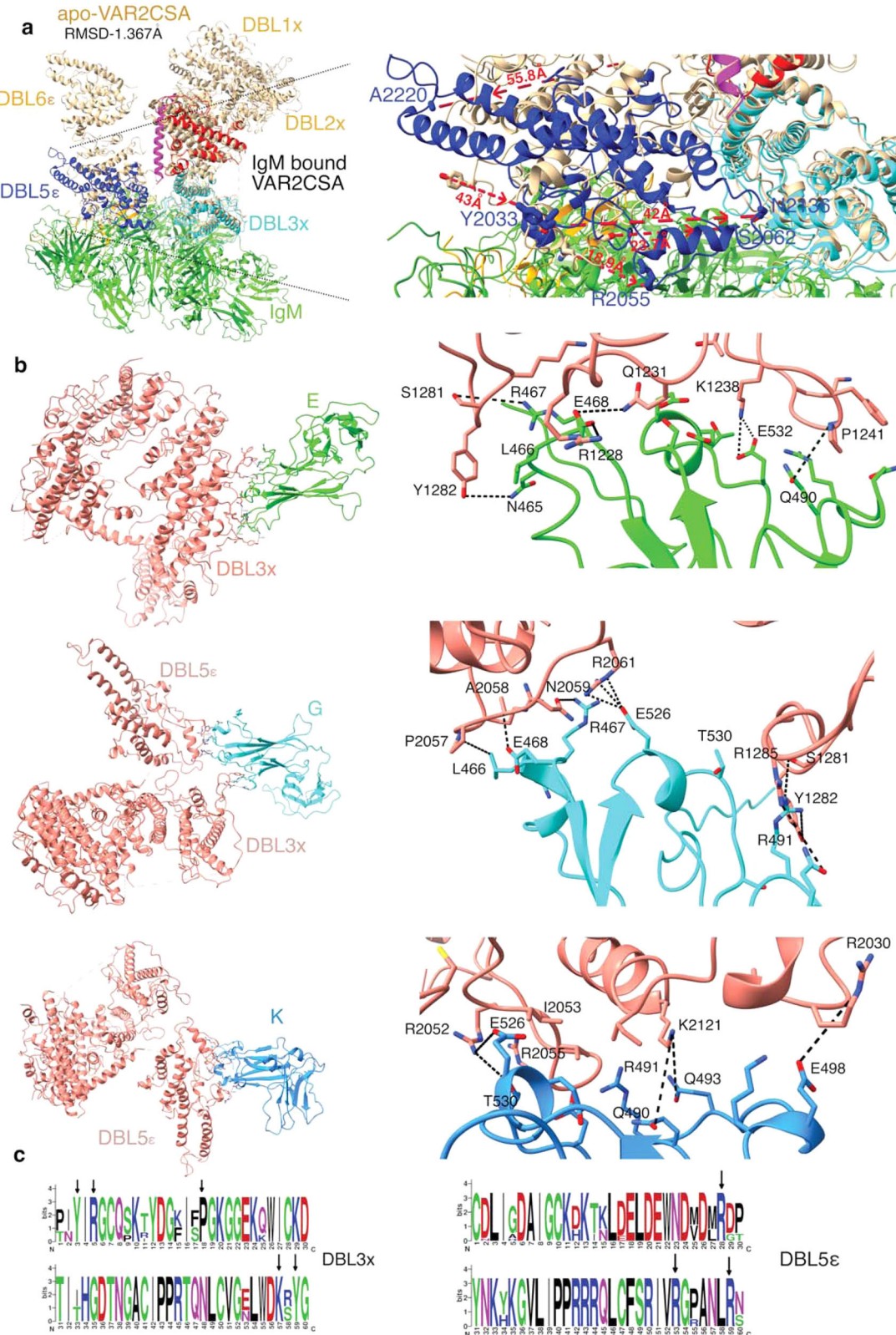

**Fig. 3 | DBL3x and DBL5ε domains interact with the planar core of IgM.**
**a** Comparison of apo-VAR2CSA(pdb7B52+pdb7NNH) and VAR2CSA in the complex (left). apo-VAR2CSA (beige). For the complex, IgM is colored green, VAR2CSA domains from Fig. 2b: ID2- red, DBL3x-cyan, ID3-magenta, DBL4ε-brown, DBL5ε-blue. The right panel shows shift of DBL5ε domain upon binding to IgM. The dashed red line shows the distances between corresponding residues of DBL5ε in apo-VAR2CSA and IgM-VAR2CSA. The arrow on the dashed red line represents the

direction of movement of residues in the complex with respect to apo-VAR2CSA. **b** Interacting interfaces between chain I (VAR2CSA) and chain E (Top), chain G (middle) and chain K(Bottom) of IgM along with respective atomic view of the interfacing residues in VAR2CSA and IgM (right panel). **c** WebLogo obtained after alignment of DBL3x and DBL5ε domains of *P. falciparum* isolates (Supplementary Fig. 8 and 9). The arrow indicates the junction forming amino acids from DBL3x and DBL5ε domains.

and chain M of VAR2CSA with a total of 271 Å$^2$ interfacing area (Supplementary Fig. 7b). PISA also suggests an interaction between chain F and chain I, (82 Å$^2$) and chain C and chain M (-30 Å$^2$). However, the buried/interfacing area is not very significant.

Since VAR2CSA is exported on the surface of IEs, it is under constant immunological pressure and as a result amino acid sequence polymorphisms have been reported in various domains of VAR2CSA[27–30]. Therefore, we checked if key residues are conserved among *P. falciparum* isolates. Our in silico analysis of DBL3X and DBL5ε sequences from many isolates confirmed conservation of key residues for the IgM-VAR2CSA interaction (Fig. 3c, Supplementary Fig. 8 and 9).

## IgM is responsible for reduced binding of VAR2CSA-expressing IEs to its receptor

In our complex, we could not model Cμ1-Cμ2 domains, however in order to understand how these domains of IgM would be placed when bound to VAR2CSA, we resolved a low resolution map of the IgM-VAR2CSA complex which revealed that Cμ1-Cμ2 domain of IgM instead of being planar like Cμ4, bends around VAR2CSA and its Cμ1-Cμ2 domains would alternate up and down from the Cμ4 plane that could cover part of VAR2CSA and impact the binding of VAR2CSA to CSA (Fig. 4a). To support this observation, we compared the surface detection of VAR2CSA in *P. falciparum* strain, CS2 cultured in either NHI-plasma (+IgM) or albumax (-IgM) by fluorescence-activated cell sorting (FACS) and observed less staining of VAR2CSA on CS2 IEs in NHI-plasma compared with albumax, suggesting the reduced accessibility of VAR2CSA on IE surface when bound to IgM (Fig. 4b). This correlates with the observed reduction in binding when CS2 is grown in NHI-plasma (Fig. 1a, b). As NHI-plasma also consist of different serum proteins, we tested if the compromised binding of CS2 IEs in NHI-plasma could be completely attributed to IgM. We added IgM to CS2 cultured in albumax media at concentrations of 10 nM, 100 nM (similar to 10% NHI-plasma) and 1000 nM (physiological levels in NHI-plasma) and tested the effect on CSA binding. We did not observe significant reduction in binding to CSA in presence of 10 nM IgM (Fig. 4c). However, binding of IEs in 100 nM and 1000 nM IgM showed ~50% reduction that is similar to NHI-plasma, confirming the steric hinderance in VAR2CSA staining (Fig. 4c). We also observed that addition of IgM at 100 nM and 1000 nM concentration to CS2 grown in albumax causes inefficient staining with VAR2CSA antibodies on IE surface, similar to NHI-plasma (Fig. 4b), establishing that IgM impairs the adhesion of VAR2CSA-positive IEs to its receptor in vtiro (Fig. 4c).

## Discussion

IgM, a soluble serum protein in humans, has been shown to play a role in the protection of parasites against phagocytosis[31]. Moreover, binding of PfEMP1s to IgM is well documented in cases of rosetting parasites, where IgM leads to clustering of PfEMP1s, such as IT4var60 and HB3var06, and causes increased rosetting[32,33]. Various studies have shown through protein-protein interaction assays that more than one domain could be involved in this interaction[21,22]. Further, it was also hitherto unclear whether more than one VAR2CSA binds to IgM, as with other PfEMP1s that mediate rosetting[33]. In our study, we resolved the map of the IgM-VAR2CSA complex and built an atomic model where the domain DBL3X and DBL5ε interact with IgM and occupies both Cμ4 faces of IgM (Fig. 2). In our complex we added more than two-fold molar excess of VAR2CSA than IgM and most of the complexes had one IgM bound to two VAR2CSA. There were just over 5000 particles that formed complex in 1:1 ratio as compared to over 1.28 million particles forming 1:2 ratio complex (Supplementary Fig. 3). More interestingly in both cases, the VAR2CSA domain that interacted with IgM are same. We also observed fractions (fractions 1 and 2 from the GraFix gradient; Fig. 1d) migrating higher than the 1:2 ratio, but we did not see any higher order of interaction; these higher fractions contained a mix of 1:2 complexes and mild aggregates. Since VAR2CSA

interacts with the Cμ4 of IgM on opposite faces, it is unlikely that more than two VAR2CSA molecules can interact with one IgM as both binding sites are occupied.

In order to get a clear picture of how two VAR2CSA are oriented to each other on opposite faces of IgM, we took three point into consideration to calculate angular separation. We chose Leu$^{566}$ on chain E of IgM as pivot point with Leu$^{2216}$ in the loop region of DBL5ε on both VAR2CSA, which shows the angular separation of 70.1°. However, angular separation between ordered alpha helix residue Lys$^{2209}$ in DBL5ε in chain I and chain M is 78° (Fig. 4d). Both angular separations are partly augmented due to the ~61° separation between two neighboring Cμ4 peptides that is separated by J chain in pentameric IgM (angle between peptide A and L separated by the J chain, Fig. 4d). This causes the C-termini of the two VAR2CSA molecules to point in the same direction, separated by 15.7 nm, towards the J chain (Fig. 4d). The distance between juxtaposed C-termini of the VAR2CSA molecules, correlated well with single molecule fluorescence measurements which found that neighboring VAR2CSA on IEs are separated by an average 14–18 nm distance[34]. This also suggests that although it is beyond the reach of monomeric IgG to bind to two VAR2CSA together, the Cμ4 of IgM that has the dimension of 18.2 nm could couple two neighboring VAR2CSA on the IE surface (Fig. 2c). Nevertheless, it is difficult to ascertain if the two neighboring VAR2CSA would always bind to an IgM on IE surface. Further, we ruled out the possibility of the VAR2CSA molecules coming from two IEs because, in such a case, their C-termini would point in opposite directions and the distance between their C-termini would be greater than 18.2 nm (Fig. 4e). A similar coupling model for the epidermal growth factor receptor (EGFR) on cell membranes is very well known and studied in the presence of its ligand EGF[35,36]. Recently Ji et al., have shown that IgM-VAR2CSA complex exists in 1:1 ratio[37]. Since, IgM used in the complex formation was a truncated construct that contained only Cμ3-Cμ4 domains, it may have artificially restricted the assembly at a lower stoichiometry. However, more importantly the interfacing residues between IgM and VAR2CSA remained same as our DBL3X and DBL5ε domains[37].

It is evident from our low resolution IgM-VAR2CSA complex map that many parts of VAR2CSA could become inaccessible to bind to CSA upon binding with IgM (Fig. 4a), We resorted to functional experiments to confirm our observation of reduced binding to CSA in CS2 cultured with NHI-plasma when compared with culture in albumax. Further, this is supported by a similar reduction of binding when we added IgM to CS2 cultured in albumax media (Fig. 4c). Our structural observation also correlates with the FACS surface detection of VAR2CSA, in which CS2 cultured in plasma (+IgM) and when IgM is added to CS2 grown in albumax media showed less staining as compared with parasites cultured in albumax (-IgM, Fig. 4b). Previously, it was also suggested that VAR2CSA on parasites is masked upon binding to IgM. However, they did not observe a reduction in CSA binding upon adding 10 nM nonimmune IgM[31]. We speculate that the low concentrations of IgM used (equivalent to IgM in 1% plasma) could be the reason for this since the concentration of IgM in 10% plasma is 100 nM and the physiological concentration is 1000 nM[38].

Interestingly, the reduced binding of VAR2CSA to its CSA receptor in +IgM media (Fig. 4c) is in complete contrast to how IgM impacts rosetting parasites[39]. Multiple studies have established that IgM binding in rosetting parasites displays more virulence as compared with nonbinders. Cryo-electron tomography of PfEMP1 (IT4VAR60), which is important for rosetting, in complex with IgM has shown that IgM wraps around IT4VAR60 and clusters PfEMP1 on IE surfaces[33]. In the case of the IgM-IT4VAR60 complex, IgM is responsible for increasing the rosetting by augmenting the binding strength of parasites to its receptor through clustering[33]. In contrast we propose that IgM binds with VAR2CSA molecules and creates a steric hinderance for VAR2CSA molecules on IEs from binding to its receptor efficiently, though this still requires in vivo demonstration (Fig. 4f).

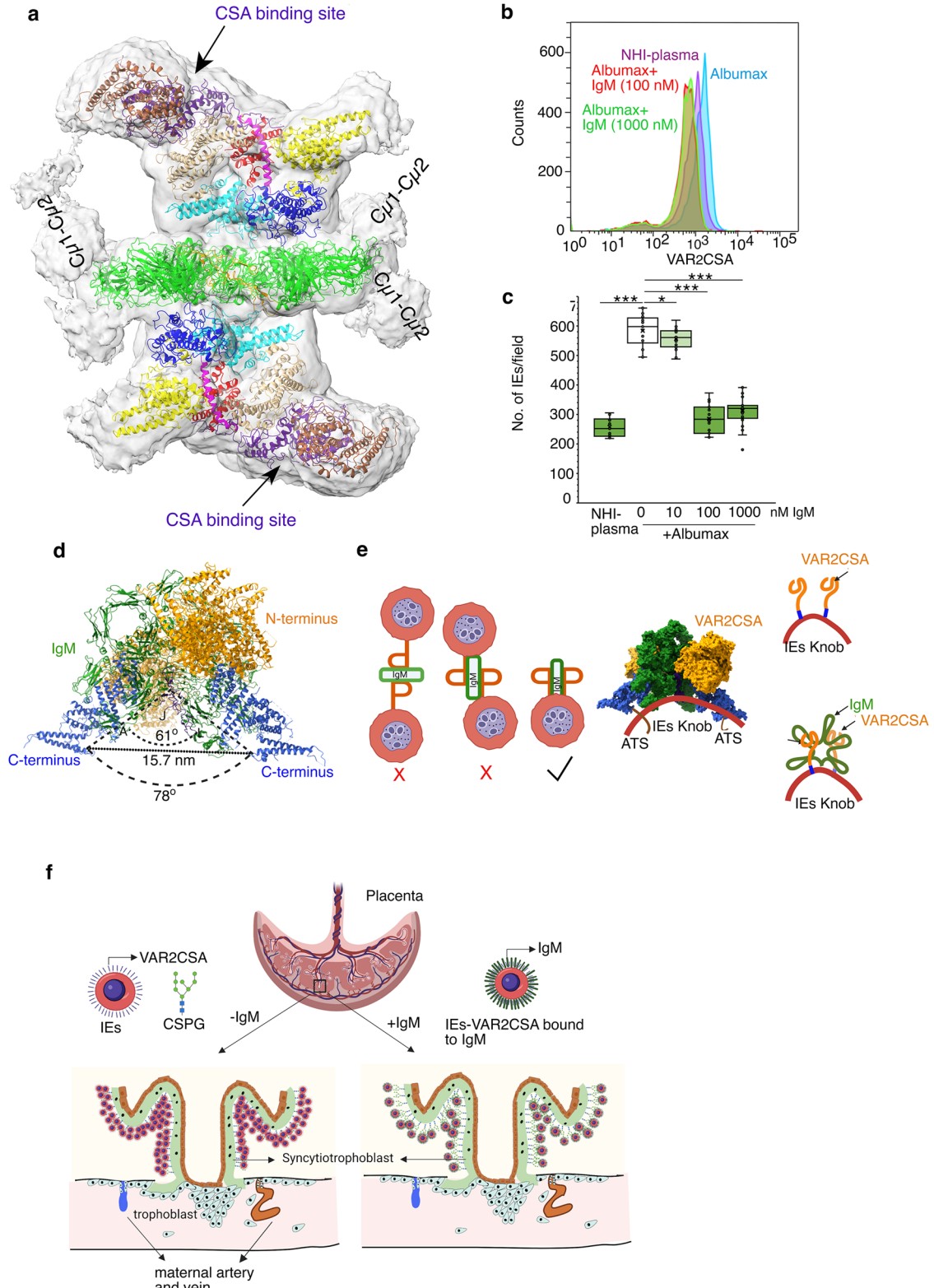

Overall, our cryo-EM study of the IgM-VAR2CSA complex demonstrates that IgM has a negative or inhibitory impact on the binding ability of *P. falciparum* IEs to its receptor in vitro.

## Methods

### Expression of VAR2CSA in Schneider 2 cells (S2)

Codon-optimized FCR3 VAR2CSA (1–2648 amino acid) was cloned into pMTBipV5HisA (Thermo Fisher Scientific, Waltham, MA). The clones were then transfected into S2 *Drosophila* cells[33]. S2 cells were maintained in log phase in Schneider media with 10% fetal calf serum (FCS). The cells were transfected following the manufacturer's protocol (Invitrogen, Carlsbad, CA). Briefly, $1 \times 10^6$ cells were plated into 24-well plates. The transfection mix was prepared by adding a mix of plasmid (19 μg), 2 M CaCl$_2$ (36 μL), and pCoBlast plasmid (1 μg), adjusted to 300 μL, mixed with an equal volume of 2× HEPES-buffered saline (50 mM HEPES, 1.5 mM Na$_2$HPO$_4$, 280 mM NaCl (pH 7.1), and incubated

**Fig. 4 | IgM protects against severe placental sequestration. a** A low resolution map of IgM-VAR2CSA complex shows the bending of Cμ1-Cμ2 domains of IgM around VAR2CSA. The map is fitted with IgM-green, VAR2CSA domains; DBL1x-brown, ID2- red, DBL3x-cyan, DBL4ε-beige, ID3-magenta DBL5ε-blue, DBL6ε-yellow. Arrow and indigo color represents CSA binding pocket in the DBL2x domain. **b** FACS analysis using anti-VAR2CSA antibody on CS2 cultured in NHI-plasma (purple), Albumax (blue). CS2 cultured in Albumax was supplemented with 100 nM (red) and 1000 nM (green) IgM. **c** The box-whisker plot of the bound CS2 IEs on CSA in NHI-plasma (green), albumax (white), albumax+IgM (10 nM, light green), albumax+IgM (100 nM, green) and albumax+IgM (1000 nm, green) were counted and plotted as number of IEs bound/field. *n* = 3 performed in duplicates and 5 fields were counted per spot and represented as a point in the plot, cross (x) represents the mean, line crossing the box plot is the median, lowest and highest whisker represents minimum and maximum data point, whereas lower bound of the box represents Quartile 1 and upper bound of the box represents Quartile 3, dot outside the whisker are outliers (original values are available in source data). *P* values were calculated using paired t-test, \*\*\**P* < 0.001, \**P* < 0.0517. **d** Atomic model (left) depicting the angle between chain A and chain L separated by J chain (bottom). green - IgM, purple-J chain, orange - N-terminus of VAR2CSA, blue - C-terminus of VAR2CSA **e** Schematic diagram of possible positions of C-termini of VAR2CSA in complex with IgM on IE surface where third option fulfills the distance and orientation criteria of (**d**). The surface rendered composite model of IgM-VAR2CSA complex represented on IE surface along with the right panel shows the schematics of presentation of VAR2CSA in absence and presence of IgM. **f** A hypothetical model for placental sequestration of IEs in presence and absence of IgM, which requires in vivo demonstration. In the presence of IgM, binding of VAR2CSA expressing IEs to CSPG (lime green) is compromised leading to lower sequestration. Panel 4 **f** was made using BioRender.

at room temperature (RT) for 30–40 min. The transfection mix was placed onto the S2 cells, and the complete S2 media was added after 24 h. After 48 h, blasticidin-HCl (25 μg/mL) was added to the transfected S2 cells. Once the S2 cells were stable in the blasticidin-HCl supplemented media, they were induced with $CuSO_4$, and the supernatant was tested for expression of VAR2CSA using a penta-Histag antibody (Cat# 34660, Qiagen). For large scale overexpression, cells were proliferated to 2 L media to $5–8 \times 10^6$ cells/mL and then transferred to Schneider's media supplemented with 0.6 μM $CuSO_4$. Media was harvested after 72 h of overexpression. The spent media was extensively dialyzed against HEPES buffered saline (HBS; 20 mM HEPES pH 7.4, 200 mM NaCl).

### Purification of VAR2CSA and preparation of the IgM-VAR2CSA complex

The dialyzed, spent media was loaded onto a Talon column that had been pre-equilibrated with HBS + 10 mM Imidazole. The column was washed with 10 column volumes (CV) of HBS + 40 mM imidazole. VAR2CSA was eluted with 5 CV of HBS + 200 mM imidazole. Fractions were checked by SDS-PAGE (Fig. 1). Later fractions were pooled, concentrated, and buffer exchanged in HBS using 100 kDa amicon devices (Millipore, Burlington, MA). Protein concentration was estimated using Bradford assay, and IgM (Jackson Immunoresearch Cat# 009-000-012) was dialyzed with HBS, mixed with VAR2CSA in a 1:2 molar ratio, and incubated at RT for 20 min. The complex was purified using SEC Superose6. The eluted peak was concentrated and layered on top of 10–40% sucrose solution gradient supplemented with 0–0.2% glutaraldehyde[40]. GraFix was run at 38,000 rpm using a SW60Ti rotor for 15 h at 4 °C. The density gradient was fractionated into 250 μL aliquots. Each fraction was checked by native-PAGE, and the best fractions were checked by negative-stain EM, concentrated, and plunge frozen using a Vitrobot into liquid ethane to prepare vitreous ice for cryo-EM data collection.

### Cryo-EM sample preparation

Cu R2/1, 200 mesh grids from Quantifoil (Jene, Germany) were used. The grids were glow discharged using a Solarus plasma cleaner (Gatan, Pleasanton, CA) for 40 s with $H_2$ and $O_2$ gas flow. For plunge freezing, 4 μL of complex was incubated on the grid for 1 min at 4 °C in 100% humidity. The sample was blotted for 4 s and plunged into liquid ethane. Grids were stored in liquid nitrogen until data collection.

### Data collection

Data were collected on a Titan Krios electron microscope equipped with a Falcon 3 camera (both from Thermo Fisher Scientific) at 300 keV (Supplementary Fig. 3). Movies were recorded using EPU software (Thermo Fisher Scientific) at a nominal magnification of 75,000× in counting mode, a pixel size of 1.1 Å at the specimen level, and with a dose rate of 0.98 e/Å²/s, which corresponded to 0.81 e/px²/s at the specimen level. The exposure time was 63.0 s, resulting in an accumulated dose of 51 e/Å[41]. Each movie included 50 fractioned frames.

### Data processing

Data were processed using RELION3.1. A total of 15,668 movies were aligned using Motioncorr2 with dose weighting[41]. Contrast transfer function estimation was done using gctf with dose weighting and equiphase averaging (Gctf-v1.18 https://www2.mrc-lmb.cam.ac.uk/research/locally-developed-software/zhang-software/)[42]. Summed images were screened to remove low-quality images. A total of 2,000 particles were manually picked to generate initial 2D classed for automatic particle picking. A total of 1,993,285 particles were auto-picked. New 2D class averages were generated on binning 2 extracts (Supplementary Fig. 3). Multiple rounds of 2D averaging were done to remove bad classes (Supplementary Fig. 3). An initial model was built using 1,289,909 particles (Supplementary Fig. 3), and further 3D classes were generated (Supplementary Fig. 3). Multiple rounds of 3D classification were performed to remove bad classes, and 3D-autorefine was performed without binning and within a circular mask of 350 Å[43]. Particles were then selected based on a "rlnNrOfSignificantSamples" value of <20 and refined within a mask. The selected particles were used for 3D-autorefine with C1 symmetry, resulting in a map at 3.6 Å resolution. A further unsharpened map or map without post-processing was used. All iso-electron potential surfaces were visualized using ChimeraX. Local resolution of the map was estimated using RELION Localres (Supplementary Fig. 3).

### Model building and refinement

In order to build an atomic model, three PDB models were used (6KXS, 7B52, and 7NNH). Receptor domains and glycosylation were removed from the original 6KXS. DBL1X and DBL2X domains were cropped from 7B52. Also, the DBL6ε region of the 7NNH was removed. All components were individually docked into the map using the "fit-in the map" function of ChimeraX[44,45] and manually corrected using COOT 0.98. The majority of manual and automated fitting and rebuilding was done using COOT 0.98[46,47] (Supplementary Fig. 10). The model was stitched together using Chimera and refined using real-space refinement in package PHENIX (V1.20.1)[48,49]. Validation was done using the PHENIX (V1.20.1) statistics shown in Supplementary Table 1. Figures for publication were generated using PyMOL (v.2.1, https://pymol.org/2/).

Structural and map figures were prepared in ChimeraX (v.1.0, https://www.rbvi.ucsf.edu/chimerax/), which are developed by UCSF, and PyMOL (v.2.1, https://pymol.org/2/).

### RMSD calculation

Two pdbs were compared using chimeraX matchmaker for c-alpha only with the iteration cuttoff of 2 Å for pruning residues. For the sequence pair alignment during RMSD calculation matchmaker used needleman-wunsch algorithm. The resullts were plotted as the gradient of RMSD

in various parts of the molecule by rendering by attribute: residue vs seq_RMSD.

## Parasite culture and CSA adhesion of parasites

CS2 parasites obtained from MR4 were cultured using standard procedures in Roswell Park Memorial Institute (RPMI) media with 10% human plasma or 5 g/L albumax and O$^+$ erythrocytes at 37 °C under mixed gas (5% $CO_2$ + 5% $O_2$ and a balance of $N_2$).

For the adhesion assay, CSA was coated at a concentration of 100 μg/mL as circular spots on plastic petri dishes overnight at 4 °C and blocked with 2% bovine serum albumin (BSA) at 37 °C for 2 h. CS2 cultured in either plasma or albumax were adjusted to equal parasitemia (~10%, $2 \times 10^6$ IEs) possessing a hematocrit of 10% and washed three times with phosphate buffered saline (PBS). The washed cultures were overlaid on the CSA-coated spots for 1 h at RT. After incubation, the petridish was washed with PBS to remove unbound parasites, and the bound IEs were fixed with 2% glutaraldehyde. The fixed parasites were stained with acridine orange and images were taken using a bright field microscope and a confocal microscope (Leica, Schott, Germany) with a 488 nm filter and 20X objective lens. The bright field and fluorescence images were merged and counted to analyze IEs bound to CSA. The data is represented as a box whisker plot. The experiment was performed 3 times in duplicates and 5 fields were counted per spot and represented as a point in the plot.

## Immunization of rabbits to raise antibodies against VAR2CSA

For immunization, New Zealand white rabbits 12 weeks old were used and approved by IAEC, SRM Chennai. The pre-bleed was collected 1 day prior to immunization. On day 0, a primary injection containing an emulsion of 200 μg of purified VAR2CSA and an equal volume of complete Freund's adjuvant (Cat#-F5881, Sigma-Aldrich, St Louis, MO) was given to the rabbit. The first, second, and third boosters, given on days 14, 42, and 70, respectively, comprised an emulsion that contained 100 μg of purified VAR2CSA and an equal volume of incomplete Freund's adjuvant(Cat#-F5506, Sigma-Aldrich, St Louis, MO). The terminal bleed was collected on day 84 and used for IgG purification.

For IgG purification, the serum was loaded on pre equilibrated Protein A column with PBS. The column was washed with PBS and the bound IgG was eluted with 0.2 M glycine (pH 2.3) and quenched with 1 M Tris-Cl. The fractions were analyzed by reducing SDS-PAGE, and the fractions containing IgG were pooled, dialyzed in PBS, and concentrated to 10 mg/mL.

The purified antibodies were tested for recognition of VAR2CSA in CS2 parasite using western blot analysis. For western blot, the CS2 parasite culture at a parasitemia of 15−20% was lysed using saponin followed by extraction with 0.5% Triton X-100. The pellet obtained after extraction was treated with 2% SDS for SDS soluble proteins. The 2% SDS extract was centrifuged at 10,000 rpm for 5 min and the supernatant was collected and bradford analysis was performed for protein quantification. 20 μg of protein was loaded and resolved on 4−12% Novex gel (Thermofisher scientific) and transferred onto nitrocellulose membrane at 70 V for 2 h. The membrane was blocked with 5% skimmed milk and washed with 1X Phosphate buffered saline +0.05% Tween-20 (PBST). Anti-VAR2CSA antibody (10 μg/ml) and Pfhsp-70 (1:1000, Cat# SPC186, Stressmarq) raised in rabbit were added for 2 h at room temperature. The blot was washed thrice with PBST and anti-rabbit IgG-HRP from donkey (1:5000, Cytiva, NA934-1ml). After washing thrice with PBST, the blot was developed with chemiluminescence reagent (Cat#-35065, Thermofisher Scientific) and visualized using chemiluminescence imaging (Amersham).

For western with anti J chain (Cat#-PA5-83707, Thermofisher Scientific) dilution of 1:1000, goat anti-human IgM heavy chain (Cat# A24484, Invitrogen) dilution of 1:1000 were used. The secondary antibodies anti-goat HRP (Cat# A15999, Thermofisher scientific) and anti-mouse IgG HRP from sheep (Cat#NA931-1ml, Cytiva) were used at a dilution of 1:5000.

## FACS analysis of parasites

The anti-VAR2CSA antibody was preabsorbed with uninfected red blood cells (RBCs) overnight at 4 °C. The CS2 culture at 10% parasitemia was washed thrice with PBS and incubated with pre-absorbed anti-VAR2CSA antibody and nonimmune IgG from rabbit at a concentration of 100 μg/mL in PBS + 2% BSA for 1 h at RT. The parasites were washed four times with PBS and incubated with 1:100 diluted anti-rabbit Alexa Fluor 594 and 2.5 μg/mL ethidium bromide for 1 h at RT. As a negative control, the parasites were only incubated with anti-rabbit Alexa Fluor 594 (1:100, Cat# A−21207, Thermofisher Scientific)and 2.5 μg/mL ethidium bromide. The parasites were washed four times with PBS and finally resuspended in PBS for FACS analysis using FACS-Celesta (Becton, Dickinson and Company, Franklin Lakes, NJ). For acquisition, unstained cells were used to gate RBCs, while ethidium bromide staining was used to gate IEs and only secondary antibody was used to adjust the gate for the positive staining with anti-VAR2CSA antibody. The VAR2CSA positive cells were tested from 10,000 IEs, and the data were analyzed using FlowJo v10.9 software, where expression of VAR2CSA was compared with CS2 grown in albumax and plasma containing media for 10,000 IEs (Fig. 4b).

## Immunofluorescence assay

CS2 culture at 10% parasitemia was washed thrice with PBS and incubated with pre-absorbed anti-VAR2CSA antibody at a concentration of 100 μg/mL in PBS + 2% BSA for 1 h at RT. The culture was then washed four times with PBS and incubated with donkey anti-rabbit Alexa Fluor 594 (1:100; Cat# A-21207, Thermo Fisher Scientific) and 6-diamidino-2-phenylindole (DAPI, 100 μg/mL) in PBS + 2% BSA for 1 h at RT. After washing, the IEs were visualized and analyzed using a confocal microscope (Leica Biosystems,) at 63× objective LASX softwaree lens.

## Statistical analysis

Results are represented as box-whisker plot with mean, median line and maxima and minima values. Paired t-test was used to calculate the statistical difference among experimental groups. Statistically significant was defined as $P < 0.05$.

## Reporting summary

Further information on research design is available in the Nature Portfolio Reporting Summary linked to this article.

## Data availability

The cryo-EM map has been deposited in the Electron Microscopy Data Bank (EMDB) under accession code EMD-34399 (IgM-VAR2CSA complex). The atomic coordinates have been deposited in the Protein Data Bank (PDB) under accession code 8GZN (IgM-VAR2CSA complex). Previously published structures can be accessed via 7B52, 7NNH, 6KXS. Raw data in this study are available upon request. Source data are provided with this paper.

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

## Acknowledgements

A core subsidy from the Okinawa Institute of Science and Technology Graduate University funded the work in the SCB unit. SG is funded by DST-ECR/2017/001703, DBT-BT/PR40722/MED/29/1529/2020, and IISER Tirupati core funding. We appreciate the support from the Scientific Computing and Data Analysis Section (OIST RSD) for the use of cluster computing. We acknowledge the Scientific Imaging Section (OIST IMG) for use of the cryo-EM facility. We also acknowledge Bijayeeta Deb, Adrita Das, and Ramya Vilvadrinath from IISER Tirupati for their help in acquiring confocal microscopy images.

## Author contributions

S.G. and U.S. are equal senior authors. R.R.A. conceived study. R.R.A. and S.G. conducted experiments, R.R.A. and S.G. analyzed data. R.R.A., S.G., and U.S. discussed results. R.R.A. wrote the manuscript with input from S.G. and U.S. All authors read and approved the manuscript.

## Competing interests

The authors declare no competing interests.
