## [Peer Review File · Nature Communications]

Cryo-electron microscopy of IgM-VAR2CSA complex reveals IgM inhibits binding of Plasmodium falciparum to Chondroitin Sulfate AReviewers' Comments:

Reviewer #1:

Remarks to the Author:

Drs. Akhouri, Goel and Skoglund describe results of their investigation of the role of nonimmune (Fc-mediated) binding of IgM to VAR2CSA-type PfEMP1, based mainly on cryo-EM data on complexes of recombinant VAR2CSA and IgM. The authors report that the interaction between the parasite ligand and the antibody occurs at 2:1 ratio and involves the DBL3 and DBL5 domains in the former and Fc μ 3-4 in the latter. According to the data presented, this results in an IgM molecule sandwiched between two VAR2CSA molecules. They furthermore report that the interaction compromises the ability of VAR2CSA-positive infected erythrocytes (IEs) to adhere to the cognate VAR2CSA receptor CSA and conclude that it therefore plays "an important role in modulating binding of var2CSA expressing parasites to host receptors and could regulate level of placental sequestration in patients". While the manuscript includes information of substantial interest and novelty, it also contains a number of flaws that must be addressed.

Major points:

1. The source of the IgM used to generate complexes with recombinant VAR2CSA should be specified.
2. The reference list should be thoroughly vetted. Many places, the authors provide three or more references, where one or two pertinent ones would do. Also, In some cases, key primary references are omitted in favour of follow-up studies and/or studies that do not address the point being made.
3. "However, IgM also binds... (heat-inactivated plasma)". This statement is factually incorrect. Barford et al. (2011) [ref. 26 in the manuscript] tested the effect of purified nonimmune IgM-binding on the ability of VAR2CSA-positive infected erythrocytes (IEs) to adhere to CSA and found that it did not inhibit adhesion.
4. The authors used Plasmodium falciparum CS2 parasites, "known to efficiently express var2CSA". The authors should document the presence of VAR2CSA on the surface of the IEs used in the experiments, rather than relying on the a >10 years old reference from another laboratory (a reference where the claim of VAR2CSA expression is itself based on a literature reference to another paper published a further decade earlier). This is an important point, as the data in Fig. 1 depend heavily on how many of the IEs used expressed VAR2CSA rather than some other PfEMP1. authors data.
 - 5.1. Fig. 1: The data shown in panel a are clearly inadequate to document the claimed IgM-mediated inhibition of IE adhesion to CSA. The low magnification makes it impossible to distinguish IEs from uninfected erythrocytes. Higher magnification and the use of a fluorescent dye to label IEs (e.g. a nuclear dye such as ethidium bromide) are needed.
 - 5.2. The number of IEs added to the assay should be stated and the proportion of IEs that were VAR2CSA-positive should be documented to enable evaluation of how many of the IEs added adhered to CSA under optimal conditions (maximum adhesion). It is a concern that panel a and b of this figure do not seem to correspond: if only the arrowed cells in panel a are IEs, then no IgM-dependent reduction in adhesion is discernible. If, in contrast, all the cells are IEs, the reduction appears much more substantial than shown in panel b, not least considering the narrow error bars (what do they represent?).
 - 5.3. Fig. 1c: A substantial fraction of the VAR2CSA in the IgM+var2CSA experiment appears free of IgM. If the interaction has high affinity, as has been reported, this is surprising unless VAR2CSA was added in substantial excess. Was that the case, and if so, why?

5.4. Fig. 1d: The gels shown are insufficient. Western blots with VAR2CSA- and IgM-specific reagents would be needed to adequately document which bands correspond to IgM, VAR2CSA, and IgM-VAR2CSA complex.

5.5. Fig. S1a is not needed, as there is no reason why IgM would be present in Albumax medium. Fig. S1b needs a loading control to document that the same amount of material was loaded in the two lanes.

6. Many of the findings reported in the current manuscript are at variance with previous studies, not least with the recent report by Ji et al. (data available at Biorxiv sine June 2022 - <https://www.biorxiv.org/content/10.1101/2022.08.03.502706v1.full>). The authors must discuss their own findings in the context of the findings of Ji et al., as several key observations are quite discordant (e.g., the stoichiometry, the impact on binding to CSA).

7. I am greatly puzzled by the apparent anti-parallel orientation of the two VAR2CSA molecules on either side of IgM in most figures. Wouldn't this be expected to lead to agglutination? The authors discuss why the orientation is in fact not anti-parallel (although it is shown that way in all figure panels but the low-resolution Fig. 4a). However, I admit not being able to follow their argument, and would encourage a more accessible explanation (and a discussion why their model rather than the model of Ji et al. has the correct stoichiometry).

8. "Since overall architecture... core of IgM": A 2:1 PfEMP1:IgM ratio has previously been reported for another PfEMP1, based on ultracentrifugation data (Stevenson et al. Cell Microbiol 2015). Perhaps a similar approach could shed light on the discrepancy between the present findings and those of Ji et al. (?). Also, it would perhaps be of interest to check whether the weblogo in DBL3 and DBL5 of VAR2CSA is also present in the non-VAR2CSA PfEMP1 variants used in previous studies of non-immune binding of IgM to PfEMP1 (e.g., several studies from the Rowe and the Hviid laboratories).

9. Fig. 4: The authors argue that VAR2CSA expression on the IE surface does not depend on whether parasites were grown in serum-containing or serum-free (Albumax) medium (Fig. S1b – but see the reservation regarding the documentation in comment 5.5). The data in Fig. 4 is therefore argued to document shielding of antibody epitopes by IgM. It would arguably be superior to compare IEs from Albumax cultures after pre-incubation with or without non-specific purified IgM (as was done previously by others, e.g., in the above-mentioned study by Barfod et al.). Why was that not done? It would have excluded uncertainties regarding exact maturation stage of the parasites employed (which markedly influences expression levels). Overall, I find Fig. 4 dispensable, as it is repetitive and mainly speculative. I suggest either removing or repopulating it with more informative panels.

10. The main conclusion drawn by the authors is that non-immune binding of IgM to VAR2CSA serves to limit the clinical consequences. The clinical significance of this is difficult to appreciate, as infection in the absence of IgM clearly is not an option in vivo.

Minor points:

1. A revised manuscript should include line and page numbers.
2. Ref. 1 should be updated to the 2022 report, which has just been published.
3. References should appear in citation order (presently ref. 29 appears before refs. 26-28).
4. References need to be reformatted, e.g., to italicize latin binomial names.
5. I suggest using the standard VAR2CSA acronym throughout, rather than var2CSA, which appears

as an unfortunate compromise between the protein (VAR2CSA) and the gene encoding it (var2csc).

6. At several places, the authors refer to adhesion/sequestration of parasites. It would be more correct to refer to infected erythrocytes (pRBCs, as the authors call them).
7. "Sequestration in the brain... PAM)2,3": I suggest removing the acronym PAM (pregnancy-associated malaria). It is superfluous and less specific than the PM (placental malaria) acronym, which is what this manuscript is about. PM should therefore be used throughout.
8. "Var2CSA is >310 kDa... placenta17-19". It is by now well established that DBL3 is not involved in the binding of VAR2CSA to CSA. I suggest removal of the current ref. 19, which is not directly relevant and addition of the primary study by Khunrae et al. (J Biol Chem 2010).
9. "We derived a 3.6Å map... and S3)". There is no Fig. S3 in the manuscript.
10. "As IgM-var2CSA... IgM core": suggest either removing "reasonably" completely (too casual and without much meaning), or replace it with a more specific statement (e.g., of the resolution obtained).
11. "Two basic residues... (Fig. 3bV)": rephrase to avoid "a bunch" (too casual).
12. "Our in silico analysis... the placenta": "Fig. 3d" should probably read "Fig. 3c" (there is no Fig. 3d).
13. "These two observations... pRBC surface": "give us a clue" is too informal. Please rephrase.
14. Ref. 34 is not relevant, as it is exclusively on membrane-bound IgM (i.e., the BCR) rather than on the pentameric IgM that is the topic of the present study.

Reviewer #2:

Remarks to the Author:

Recommendation: Re-write and re-submit

I think this manuscript has the potential to be an impactful paper; however it is so poorly written and formatted that I cannot complete the review. It is simply too difficult to follow even for a CryoEM and IgM expert. I suggest authors reformat their manuscript to meet the Nature Communication Guidelines (some of which I have linked below) and work with a copy editor or a similar professional to improve language communication and then resubmit.

Writing in English language is poor and needs to be addressed throughout the manuscript. Examples:

1. In the Abstract the acronym var2CSA is not defined and it is not clear what var2CSA is (e.g. a protein, cell type. etc.).
2. "A 3.6Å cryo-EM map of IgM-var2CSA complex reveals that interlocking stacking of sheets at IgM core creates planar identical face which imposes an identical binding of two var2CSA to one IgM through its DBL3X and DBL5ε domains." Is this a structure or a map? Are these beta sheets?
3. "... parasitized RBCs (pRBCs) to its receptor" should be to their receptor.
4. Throughout the manuscript there are errors in tense, pronoun usage and many cases where the words "the" "it" "a" etc. are absent as well as important nouns. For example, ""The first low resolution SAXS was successful in providing the overall architecture of var2CSA...." SAXS should modify a term such as "structure" or "data", etc." In many cases the sentences were very hard to interpret to the extent that it was difficult to review the work.
5. Paragraph structure was also not optimal making it difficult to interpret. AND there are not clear Introduction, Results, Discussion, etc. For example in what appears to be the "introduction," the

middle of a paragraph describing background information authors add sentences beginning with." In order to test whether IgM influences the binding of var2CSA to CSPG, we compared the CSPG binding ability of CS2 parasites²⁹ (known to efficiently express var2CSA)..." This should at least be a new paragraph but this sentence and the rest of the paragraph would probably be better suited for a "Results" section.

This manuscript is not formatted according to journal guidelines. There are no page numbers, no headings (e.g. Introduction, Results, Discussion) and no Subheading. Authors should closely review:

<https://www.nature.com/ncomms/submit/how-to-submit>

and

<https://www.nature.com/ncomms/submit/article>

" The main text of an Article should begin with a section headed Introduction of referenced text that expands on the background of the work (some overlap with the abstract is acceptable), followed by sections headed Results, Discussion (if appropriate) and Methods (if appropriate). The Results and Methods sections should be divided by topical subheadings; the Discussion should be succinct and may not contain subheadings. Methods are typically less than 3000 words. Figure legends are limited to 350 words each. As a guide, references should not exceed 70. Footnotes are not used."

Response to Reviewer1:

We sincerely thank the reviewer for giving time and effort to review our manuscript to remove flaws and improve quality of the manuscript. We have rewritten the manuscript in the Nature Communications format that includes Introduction, Results, Discussion and Methods. We have also taken professional language and manuscript editing services. We hope this has improved the manuscript. We have made changes in our manuscript to incorporate suggested changes. Please find point by point response to reviewers' comments below (in blue).

Major points:

1. The source of the IgM used to generate complexes with recombinant VAR2CSA should be specified.

We have included the source of IgM in the revised Methods section: Jackson Immunoresearch cat # 009-000-012 ChromPure Human IgM (Myeloma), whole molecule (included in Methods: Line #388, page-15)

2. The reference list should be thoroughly vetted. Many places, the authors provide three or more references, where one or two pertinent ones would do. Also, In some cases, key primary references are omitted in favour of follow-up studies and/or studies that do not address the point being made.

We have tried to improve reference list and tried to include original work as much as we could. However, we could incorporate more if suggested.

3. "However, IgM also binds... (heat-inactivated plasma)". This statement is factually incorrect. Barfod et al. (2011) [ref. 26 in the manuscript] tested the effect of purified nonimmune IgM-binding on the ability of VAR2CSA-positive infected erythrocytes (IEs) to adhere to CSA and found that it did not inhibit adhesion.

We have removed this sentence from the introduction part of our manuscript. However, I would be obliged if you could please see response to comment 9 as well.

4. The authors used *Plasmodium falciparum* CS2 parasites, "known to efficiently express var2CSA". The authors should document the presence of VAR2CSA on the surface of the IEs used in the experiments, rather than relying on the a >10 years old reference from another laboratory (a reference where the claim of VAR2CSA expression is itself based on a literature reference to another paper published a further decade earlier). This is an important point, as the data in Fig. 1 depend heavily on how many of the IEs used expressed VAR2CSA rather than some other PfEMP1. authors data.

We performed live IFA to confirm expression of VAR2CSA on IEs of CS2 strain of *Plasmodium falciparum* (using anti-VAR2CSA antibodies). Results are included in Supplementary Figure. 1a. Further, we routinely selected CS2 on CSA to maintain high levels of VAR2CSA on IEs.

5.1. Fig. 1: The data shown in panel a are clearly inadequate to document the claimed IgM-mediated inhibition of IE adhesion to CSA. The low magnification makes it impossible to distinguish IEs from uninfected erythrocytes. Higher magnification and the use of a fluorescent dye to label IEs (e.g. a nuclear dye such as ethidium bromide) are needed.

We appreciate reviewers' comments that low magnification images are not clear. In order to resolve this shortcoming, we repeated CSA binding assays in NHI-plasma and albumax. We further stained the bound IEs using acridine orange (stains DNA). For sharpness of images, we used confocal microscope. We captured images of bound cells in bright field as well as 488 filter, merged both to show all bound cells were IEs (Fig. 1a).

5.2. The number of IEs added to the assay should be stated and the proportion of IEs that were VAR2CSA-positive should be documented to enable evaluation of how many of the IEs added adhered to CSA under optimal conditions (maximum adhesion). It is a concern that panel a and b of this figure do not seem to correspond: if only the arrowed cells in panel a are IEs, then no IgM-dependent reduction in adhesion is discernible. If, in contrast, all the cells are IEs, the reduction appears much more substantial than shown in panel b, not least considering the narrow error bars (what do they represent?).

We apologize for confusing figure. In the repeated experiment and revised figure, we have included more representative images. In the current figure, the data is represented as box whisker plot that has actual count per field instead of accumulated counts; the line represents the mean. The experiment was performed 3 times in duplicates and 5 fields were counted per spot. Counts of IEs per field is represented as a data point in the plot. Therefore, it has total 30 data points (Fig 1a and b).

5.3. Fig. 1c: A substantial fraction of the VAR2CSA in the IgM+var2CSA experiment appears free of IgM. If the interaction has high affinity, as has been reported, this is surprising unless VAR2CSA was added in substantial excess. Was that the case, and if so, why?

Yes, VAR2CSA is added in a bit > than 2M excess of IgM. Initial experiments when we added in 1:1 ratio, it produced heterogeneous mixture of the complex. Since, initial screening results indicated 2 VAR2CSA to 1IgM complex, we added excess of VAR2CSA to drive complex to promote complete occupation of binding sites on IgM. This produced more homogeneous 2D classes in a smaller number of movies. We have included this in page 10 line#264 and page 15 line#389.

5.4. Fig. 1d: The gels shown are insufficient. Western blots with VAR2CSA- and IgM-specific reagents would be needed to adequately document which bands correspond to IgM, VAR2CSA, and IgM-VAR2CSA complex.

To show that the complex consists of both IgM and VAR2CSA, we have performed two sets of experiments.

1. Preparative ultracentrifugation for gradient separation of complex in (a) 10-40% sucrose gradient with glutaraldehyde crosslinker and (b) 10-40% sucrose gradient without crosslinker. Fractions were resolved on Native-PAGE to show that both complexes were migrating similar. We also resolved fractions from non-cross-linked gradient in reducing SDS-PAGE to show that it has VAR2CSA, IgM light chain and IgM heavy chain (Supplementary Figure 2).
2. Since we used crosslinked fraction for cryo-EM studies, resolved, transferred fraction and detected using anti-VAR2CSA antibodies, anti-C μ 4 and anti-J chain antibodies (Fig 1d).

5.5. Fig. S1a is not needed, as there is no reason why IgM would be present in Albumax medium. Fig. S1b needs a loading control to document that the same amount of material was loaded in the two lanes.

We have removed S1a. We apologize for not including loading control for VAR2CSA. We have repeated western blot for VAR2CSA in Albumax and NHI-Plasma culture with HSP70 antibody loading control (Supplementary Figure 1b).

6. Many of the findings reported in the current manuscript are at variance with previous studies, not least with the recent report by Ji et al. (data available at Biorxiv sine June 2022 - <https://www.biorxiv.org/content/10.1101/2022.08.03.502706v1.full>). The authors must discuss their own findings in the context of the findings of Ji et al., as several key observations are quite discordant (e.g., the stoichiometry, the impact on binding to CSA).

We understand reviewers' concern how our stoichiometry differs from the Ji et al.,. However, it will be a little difficult to predict why Ji et al., have got only 1:1 ratio. One reason that we can speculate is the use of truncated IgM for complex formation. However, natural IgM may behave a bit different as it does not exist in that truncated form. We hope reviewers for the Ji et al., will raise this question from them and will ask them to verify their results with whole IgM as well.

On the suggestion of reviewer, we reanalyzed our data and obtained 2 very small class averages in 1:1 ratio that has only 4500+ particles. It is negligible in comparison to the particles in 2:1 ratio. We have included 1:1 ratio class in our revised results in Supplementary Figure 3. Also, the residues interacting in Ji et al and our results are same. This further establishes although our stoichiometry is different but the mode of interaction remains same.

7. I am greatly puzzled by the apparent anti-parallel orientation of the two VAR2CSA molecules on either side of IgM in most figures. Wouldn't this be expected to lead to agglutination? The authors discuss why the orientation is in fact not anti-parallel (although it is shown that way in all figure panels but the low-resolution Fig. 4a). However, I admit not being able to follow their argument, and would encourage a more accessible explanation (and a discussion why their model rather than the model of Ji et al. has the correct stoichiometry).

I understand reviewers' concern that we didn't include models and cartoons to explain the juxtaposition of both VAR2CSA C-termini. In our revised manuscript, we have added more schematic models to show how we ruled out that VAR2CSA on opposite surface may not come from two IEs. Further, our measurements for distance between both C-termini is similar to single molecule fluorescence data from Sanchez CP et al., 2019 (Communications Biology, ref# 53). Here, C-termini are proximal to J chain and placed closer to the planar core while N-terminal domains are protruding away from core of IgM. With the help of cartoons and models we have explained how both VAR2CSA C-termini are juxtaposed and could be placed on the same knob of IE (Fig 4c, d). This is also explained in discussion, Page 10-11, line#272-285.

8. "Since overall architecture... core of IgM": A 2:1 PfEMP1:IgM ratio has previously been reported for another PfEMP1, based on ultracentrifugation data (Stevenson et al. Cell Microbiol 2015). Perhaps a similar approach could shed light on the discrepancy between the present findings and those of Ji et al. (?). Also, it would perhaps be of interest to check whether the weblogo in DBL3 and DBL5 of VAR2CSA is also present in the non-VAR2CSA PfEMP1 variants used in previous studies of non-immune binding of IgM to PfEMP1 (e.g., several studies from the Rowe and the Hviid laboratories).

We understand reviewer's concern regarding ratio of interaction between IgM and VAR2CSA. Previously Stevansson et al., have used analytical ultracentrifugation to assess stoichiometry of interaction between HB3VAR06 and alpha 2 macroglobulin.

In our study, we needed to separate mixed species of complex to get homogeneous fraction to generate good class averages.

We have analyzed fractions 1,2 and 3 post gradient fixation and we found large amounts of aggregates (as observed through Native-PAGE and cryo-EM screening). In fraction 4 and 5 we mostly got complex in 2:1 ratio. We have re-analyzed our data and we found very few complexes in 1:1 ratio (Supplementary Fig 3). Therefore, it seems most favored ratio is 2:1. Ji et al., haven't performed gradient fixation, rather they have just fixed the complex in glutaraldehyde by directly adding it to the complex upon sizing. The discrepancy in our stoichiometry could be due to the difference in the method we followed. On top of that using truncated IgM could also be possible reason for this difference.

In the following question: We have tried build weblogo for other IgM binding domain from other PfEMP1s (IT4VAR60, HB3VAR06 or TM284VAR). Unfortunately, we did not get any conservation of motifs across different PfEMP1 in the IgM binding domains. We speculate that along with our IgM-VAR2CSA complex we need detailed structural information from one of the rosetting PfEMP1-IgM complex to get more useful weblogo.

9. Fig. 4: The authors argue that VAR2CSA expression on the IE surface does not depend on whether parasites were grown in serum-containing or serum-free (Albumax) medium (Fig. S1b – but see the reservation regarding the documentation in comment 5.5). The data in Fig. 4 is therefore argued to document shielding of antibody epitopes by IgM. It would arguably be superior to compare IEs from Albumax cultures after pre-incubation with or

without non-specific purified IgM (as was done previously by others, e.g., in the above-mentioned study by Barfod et al.). Why was that not done? It would have excluded uncertainties regarding exact maturation stage of the parasites employed (which markedly influences expression levels). Overall, I find Fig. 4 dispensable, as it is repetitive and mainly speculative. I suggest either removing or repopulating it with more informative panels.

As reviewer pointed out, we have done assay by adding IgM to albumax culture and measured extent of reduction of IEs binding ability to CSA. We took CS2 in albumax and divided in four parts (see below) which ruled out stage maturation differences. We have considered three concentrations of IgM and a control.

1. 10 nM IgM as used by Barfod et al., 2011.
2. 100 nM IgM (equivalent to 10% plasma).
3. 1,000 nM IgM (equivalent to physiological concentration)

In 10 nM concentration (equivalent to 1% plasma), our results were similar to Barfod et al., with no significant reduction in binding of IEs to CSA. However, we observed ~50 % reduction in 100 nm and 1000 nM concentration of IgM which is comparable to reduction in NHI-plasma (Fig. 4b), supporting our hypothesis of IgM masking of VAR2CSA.

Since, Barfod et al., show unaffected binding in normal human serum, we were also concerned if they used NHI-serum or HI-serum for this assay. Therefore, we also analyzed CSA binding of CS2 cultured in HI-plasma. In our results we found similar binding in HI-plasma and albumax; which Barfod et al also observed when they compared albumax with NHS (normal human serum). The results are presented below for the reviewers only. If reviewers suggest we could include it in the main manuscript.

Based on this result, we looked for specific statement in the manuscript that would suggest that the serum used in the study was not heat inactivated. We did not find such statement in the manuscript. Therefore, we would like to assume that they used heat inactivated serum (NHS).

Also, we have re-populated Fig 4 with influence of adding IgM to albumax parasite culture and testing CSA binding. Also, we have used models to explain readers about the juxtaposition of C-terminus of VAR2CSA in the IgM-VAR2CSA complex.

10. The main conclusion drawn by the authors is that non-immune binding of IgM to VAR2CSA serves to limit the clinical consequences. The clinical significance of this is difficult to appreciate, as infection in the absence of IgM clearly is not an option in vivo.

We understand that it is not possible *in vivo* for patients to have IgM-deficient plasma. However, it is just a proposition that we have made to show how existence of IgM has contributed to our advantage in placental malaria just like the existence of IgM is shown to be disadvantageous when we get infected with rosetting parasites PfEMP1 that use IgM to mediate severity. We have toned down to just speculation page 12 Line # 313-319.

Minor points:

1. A revised manuscript should include line and page numbers.

We have added page number and line number in revised manuscript.

2. Ref. 1 should be updated to the 2022 report, which has just been published.

We have revised the reference to 2022 report.

3. References should appear in citation order (presently ref. 29 appears before refs. 26-28).

We apologize for our mistake; we have fixed citation order.

4. References need to be reformatted, e.g., to italicize latin binomial names.

We have formatted the reference in revised manuscript

5. I suggest using the standard VAR2CSA acronym throughout, rather than var2CSA, which appears as an unfortunate compromise between the protein (VAR2CSA) and the gene encoding it (var2csa).

We have used VAR2CSA as suggested by the reviewer throughout the revised manuscript

6. At several places, the authors refer to adhesion/sequestration of parasites. It would be more correct to refer to infected erythrocytes (pRBCs, as the authors call them).

We have used infected erythrocytes IEs as suggested by the reviewer.

7. “Sequestration in the brain... PAM)2,3”: I suggest removing the acronym PAM (pregnancy-associated malaria). It is superfluous and less specific than the PM (placental malaria) acronym, which is what this manuscript is about. PM should therefore be used throughout.

We have used PM as suggested by the reviewer.

8. “Var2CSA is >310 kDa... placenta17-19”. It is by now well established that DBL3 is not involved in the binding of VAR2CSA to CSA. I suggest removal of the current ref. 19, which is not directly relevant and addition of the primary study by Khunrae et al. (J Biol Chem 2010).

We have removed ref 19 from the previous version of the manuscript and included the reference Khunrae et al., JMB 2010, ref #22 in the new version.

9. “We derived a 3.6Å map... and S3)”. There is no Fig. S3 in the manuscript.

We apologize for our mistake; we have removed this discrepancy.

10. “As IgM-var2CSA... IgM core”: suggest either removing “reasonably” completely (too casual and without much meaning), or replace it with a more specific statement (e.g., of the resolution obtained).

We have replaced the sentence with more specific one.

11. “Two basic residues... (Fig. 3bV)”: rephrase to avoid “a bunch” (too casual).

We have made changes as suggested by the reviewer in the revised manuscript.

12. “Our in silico analysis... the placenta”: “Fig. 3d” should probably read “Fig. 3c” (there is no Fig. 3d).

We apologize for the mistake. We have corrected the fig. number in revised manuscript.

13. “These two observations... pRBC surface”: “give us a clue” is too informal. Please rephrase.

We have re written the manuscript as suggested by the reviewer and this sentence is removed from the current version.

14. Ref. 34 is not relevant, as it is exclusively on membrane-bound IgM (i.e., the BCR) rather than on the pentameric IgM that is the topic of the present study.

Ref #34 in previous version of manuscript is included because we have used PDB 6KXS for model building. In the revised manuscript it is re-numbered to Ref #45. It is referred where we discuss about IgM planar core structure. Unfortunately, we cannot exclude it from the manuscript.

Reviewer #2 (Remarks to the Author):

Recommendation: Re-write and re-submit

I think this manuscript the potential to be an impactful paper; however, it is so poorly

written and formatted that I cannot complete the review. It is simply too difficult to follow even for a CryoEM and IgM expert. I suggest authors reformat their manuscript to meet the Nature Communication Guidelines (some of which I have linked below) and work with a copy editor or a similar professional to improve language communication and then resubmit.

We appreciate the reviewers' motivation to improve manuscript. We have rewritten the manuscript in the Nature Communications format that includes Introduction, Results, Discussion and Methods. We have also taken professional language and manuscript editing services. We hope this has improved the manuscript.

Writing in English language is poor and needs to be addressed throughout the manuscript. Examples:

1. In the Abstract the acronym var2CSA is not defined and it is not clear what var2CSA is (e.g. a protein, cell type. etc.).

Unfortunately, VAR2CSA is the real full name of the molecule (a protein molecule) also mentioned in Introduction page 3 line # 65. It is not an acronym. We apologize for confusing nomenclature.

We have also taken professional language and manuscript editing services. We hope this has improved the manuscript.

2. "A 3.6Å cryo-EM map of IgM-var2CSA complex reveals that interlocking stacking of sheets at IgM core creates planar identical face which imposes an identical binding of two var2CSA to one IgM through its DBL3X and DBL5ε domains." Is this a structure or a map? Are these beta sheets?

We again apologize for confusion. Usually, crystal structures are referred to as density whereas the cryo-EM generated structures are called maps (which is a short form of isoelectron potential map or isoelectron potential surface). So, the current map is a cryo-EM derived structure.

In the center of the IgM, it is stacks of beta sheets that interlock.

We have rewritten the manuscript and elaborated in the revised manuscript.

3. "... parasitized RBCs (pRBCs) to its receptor" should be to their receptor.

We have rewritten the manuscript. This sentence does not exist in the manuscript.

4. Throughout the manuscript there are errors in tense, pronoun usage and many cases where the words "the" "it" "a" etc. are absent as well as important nouns. For example, ""The first low resolution SAXS was successful in providing the overall architecture of var2CSA...." SAXS should modify a term such as "structure" or "data", etc." In many cases the sentences were very hard to interpret to the extent that it was difficult to review the work.

We have revised the manuscript for all errors now. Also, we have taken professional language and manuscript editing services. We hope this has improved the manuscript.

5. Paragraph structure was also not optimal making it difficult to interpret. AND there are not clear Introduction, Results, Discussion, etc. For example in what appears to be the "introduction," the middle of a paragraph describing background information authors add sentences beginning with." In order to test whether IgM influences the binding of var2CSA to CSPG, we compared the CSPG binding ability of CS2 parasites²⁹ (known to efficiently express var2CSA)..." This should at least be a new paragraph but this sentence and the rest of the paragraph would probably be better suited for a "Results" section.

This manuscript is not formatted according to journal guidelines. There are no page numbers, no headings (e.g. Introduction, Results, Discussion) and no Subheading. Authors should closely review:

Unfortunately, it was a transferred manuscript therefore did not have sections like Introduction, Results, Discussion and Methods.

We appreciate the reviewers' motivation to improve manuscript. We have rewritten the manuscript in the Nature Communications format that includes Introduction, Results, Discussion and Methods. We have also taken professional language and manuscript editing services. We hope this has improved the manuscript.

Reviewers' Comments:

Reviewer #1:

Remarks to the Author:

Please see attached marked-up manuscript.

In addition:

The absence of a "Track-Changes" version of the manuscript makes it quite laborious to check the changes between the original and the current manuscript.

Re. my original Major Point 4: The data provided is an improvement. However, assessment of VAR2CSA expression by flow cytometry would be much better (Supplementary Fig. 1a documents expression for just a very few IEs, leaving it completely unknown how many of the IEs in the culture actually expressed VAR2CSA.

Re. my original Major Point 5.1: The revised figure is an improvement.

Re. my original Major Point 5.2: There is still no information on how many IEs were added to the assay, only how many that stuck. To illustrate what I mean, let us assume that 100 IEs were added, and 10 bound. This would be 10% adhesion. However, if 10,000 IEs were added, the same 10 bound IEs would only represent 0.1%. Information on how many IEs were added to the assay is therefore critical to evaluate the results.

Re. my original Major Point 5.4: The blot data in the right side of Fig. 1d need labeling indicating the location and identity of the relevant bands (which are hard to see and/or appear strangely dot-like).

Re. my original Major Point 6: I find it highly disturbing that the revised manuscript remains completely silent on the Ji et al. study, despite my explicit request to discuss their highly discordant - and as far as I can see, nevertheless quite solid - data.

Re. my original Major Point 7: The study by Stevenson et al. Cell Microbiol 2015 studied the interaction between HB3VAR60 and IgM, not α 2-macroglobulin, as claimed by the authors in the response.

Re. my original Major Point 10: Still several instances, where the suggested clinical relevance of the proposed impairment adhesion of VAR2CSA-positive IEs due to IgM needs tempered (please refer to the attached marked-up manuscript).

New Major comment:

1. Please provide details regarding the "anti-VAR2CSA antibody" mentioned several times in the text.

Reviewer #2:

Remarks to the Author:

In this manuscript, authors Goel, Akhouri and Skoglund present data detailing a cryoEM structure of human IgM in complex with the Plasmodium falciparum protein complex VAR2CSA. VAR2CSA is known to contribute to placental malaria by promoting localization of P. falciparum-infected erythrocytes (IEs) to chondroitin sulfate proteoglycans. VAR2CSA is also known to bind to human immunoglobulin (Ig) M, although the functional consequences (for the host and parasite) of this interaction have remained unclear. The structure together with supporting biological experiments reveals that two VAR2CSA can bind IgM, details the IgM- VAR2CSA molecular interface and demonstrates that IgM binding to VAR2CSA limits its interaction with host factors, including its receptor and proteoglycans. Based on data, authors contend that IgM may function to "modulate the sequestration IEs on syncytiotrophoblast

surfaces." Overall, I think this work presents significant findings relevant to understanding IgM and VAR2CSA functions and more broadly, IgM interactions with parasite proteins and factors affecting the pathophysiology of *P. falciparum* infection and associated malaria disease. While the topic and data are interesting, structural analysis and presentation (e.g. figure content) of structural findings appear incomplete and need to be improved. As presented, it is difficult to evaluate if the data fully support the conclusions that authors have drawn. Additionally, I suspect that structural findings could include insights authors have not presented, and thus further attention to data analysis should not only improve, but perhaps expand upon, the conclusions presented and allow for deeper discussion. I anticipate that authors should be able to address these weaknesses in a revised manuscript; I have listed minor and major comments below.

General comments on the introduction:

Authors may want to change immunoglobulin "mu" to immunoglobulin "M". I believe there are multiple, historic ways to describe IgM, but in practice I think most often "mu" typically refers just to the heavy chain whereas "M" would refer to the entire antibody, which seems to be what is implied in cases where this term is used.

Some basic introduction to IgM would be helpful (e.g. that it contains a JC and five IgM monomers, each with two heavy chains and two light chains). This will help the audience later when anti-JC western blots are performed and when the structure is described (see specific comments below).

Major/minor Comments by line:

Line 59. Suggest changing "parasite" to *P. falciparum* and defining the PfEMP1 acronym.

Line 73. Add a space between VAR2CSA and "is"

Line 82. I am not sure if it matters if binding is occurring to immune or non-immune IgM? Both would be interesting and relevant.

Line 83. Suggest removing the word "bulky" because it's a relative description. It's also not clear if the size of IgM would matter (nor have authors introduced the size or shape of IgM at this point in the text; see comments in results on this point) in the context of VAR2CSA binding to CSA.

Lines 151-155. Authors may want to clarify that the high-resolution structure they are citing is Secretory (S) IgM, not serum IgM and thus, that published structure contains the secretory component (SC).

Line 156. Authors state that VAR2CSA interacts with the core of IgM and they say they "focus" on residues 561-568; however they do not actually provide a description of the entire VAR2CSA-IgM interface. Here starting the paragraph by highlighting all IgM domains contacted by VAR2CSA and perhaps the total buried surface area would give the audience an overview before they start to focus on the tailpieces (which I don't think form the entire interface, but the figures are so small I cannot tell). In sum, it is unclear if authors have "cherry picked" interactions to discuss or if fact they have mentioned all that are present in their structure (maybe VAR2CSA only contacts the tailpieces, maybe not).

Line 158. Here the term tailpieces is used for the first time without definition (it should be stated that these are C-terminal extensions following the Ig-fold of C μ 4). Again, as mentioned above, it would help if IgM domain organization and structure were mentioned in the introduction.

Line 167 (and 172). Authors state that VAR2CSA engages two "identical" surfaces on opposite faces of IgM. This may be true, but the statement is not justified without presenting a structural alignment and

report of RMSD of interfacing residues on both sides. While the IgM tailpieces each share the same sequences, it is well established that the JC and its binding to IgM (or IgA) introduces asymmetry into the resulting complex so it is not clear the two sides are in fact "identical" (for example does VAR2CSA contact the JC?).

Line 176-177. Authors mention that loop regions in IgM and VAR2CSA mediate the interface and cite Fig 2, but images in Fig 2 are very small and one cannot clearly see this interface.

Line 179-181. Authors cite Fig 3a and discuss membrane proximal region and N-terminal regions of VAR2CSA however the figure doesn't provide any visual reference to these regions so it is not clear what they are referring to.

Line 183. Authors note that the conformation of IgM-VAR2CSA is different than the published apo and CSA-bound VAR2CSA structures. HOWEVER, figures or RMSDs demonstrating this are not reported. This is necessary because otherwise we have no proof that their structure is different and these conformation differences could be quite interesting and relevant to their conclusions. Structural analysis of the differences among these structures should be added at least as a supplemental figure but maybe even main text would help, especially because on lines 193 and 197 authors come back to discussion of differences between their structure and the apo structure and claim there is a conformational difference but there are NO figures or any presented analysis or quantification showing that they are different. To appropriately describe a conformational difference, analysis will likely have to comment on the interfaces between VAR2CSA components and differences among reported structures containing VAR2CSA.

Line 186 and 189. The terms "stiff" helix and "thin" linker are not biophysically appropriate. Authors have not measured the stiffness of the helix nor would one expect some linkers to be "thin" while others would be "thick"???

Line 198-199 Authors state "Since we observed the formation of a new core accompanied by conformational changes;" a new core and conformational changes compared to what? I suspect authors are referring to the other structures they have cited (but not shown a comparison to); however the way this is written is not clear.

Lines 232-233. Authors state "It is highly likely that bound IgM on IEs would mask VAR2CSA owing to its bulky size and the formation of a new core structure in the IgM-VAR2CSA complex". Authors have presented no justification for this statement. As presented, it's a hypothesis and should be worded accordingly. As noted above, some discussion of the dimensions of IgM need to be introduced in the introduction or here. Authors could conduct modeling and specify potential geometry and dimensions based on modeling to support this hypothesis but that may be beyond the scope of this work.

Lines 236 -237. I think authors need to be careful about using the term "masking". Their data do not really support any number of possible mechanisms for how IgM might block VAR2CSA binding to CS2." Authors observe less staining of VAR2CSA on CS2 IEs in NHI-plasma compared with albumax; so indeed, something is limiting the interaction but whether it is a change in kinetics, the "large size" of IgM or steric occlusion of the binding site (independent of the size of IgM) on VAR2CSA has not been worked out. I am also curious if one or two VAR2CSA always bind to one IgM in vivo? How many copies of VAR2CSA might need to be bound by IgM? This circles back to my comment earlier comment wondering if the binding sites on both sides of IgM are indeed identical? Authors discuss that one or two VAR2CSA could bind later in the discussion (where I have added a few more comments on their masking hypothesis).

Line 274 The wording "VAR2CSA molecules bound to the opposite faces of IgM begin their interaction at approximately 90 apart" is confusing. The figure clarifies this some; however I would phrase this in terms of where the VAR2CSA are located as opposed to say "beginning," which implies that authors

are discussing the order or timing of a process. The structure is static and the sentence as written implies that the binding mechanism is beginning with VAR2CSA binding in one way and continuing in another way (which I don't think is what the authors want to say here). I actually think some of the information on the structure presented in this paragraph (angles of binding and location of the termini, etc.) might be better suited for the results section, where I have indicated that authors should more completely describe the overall structure. If this approach is implemented, the other contents of this paragraph can remain in the discussion to describe the biological significance of the findings.

Line 287. Authors state that IgM is larger than VAR2CSA and therefore it could mask VAR2CSA; however large size doesn't not imply masking nor do authors quantify this apparent size difference in any way. They need to do something to better describe this to the audience and describe how the masking would occur. Which domains of VAR2CSA would likely be masked? Would IgM Fabs contribute? Would IgM N-linked glycans contribute? Because the Fabs are not visible in the structure, most of their structure figures do not give the impression that IgM is larger than VAR2CSA- particularly Fig. 4c! Perhaps Authors can make a schematic illustration of the hypothesis to include in the figure?

lines 304-306. Authors are missing a reference for the cryo-ET of PfEMP1 in complex with IgM cryoET and I cannot make further judgement of the contradiction they raise without seeing the data. I think this will be important to 1) highlight the contradiction between current model and previous study and 2) support the binding model proposed in Fig. 4d. Overall the discussion could better link their data and model and therefore together with the issues noted above, it is a bit hard to be sure if their data do indeed support the model they propose.

Other comments:

Results or discussion:

Authors fail to discuss how their structure might differ from secretory (S) IgM structures, which contain the secretory component (SC) and specifically whether or not the VAR2CSA overlaps with the SC binding site. Although it's unlikely that SIgM and VAR2CSA would ever cross paths (SIgM is located in mucosa, not blood), it's possible VAR2CSA evolved to bind at sites overlapping with SC, which would be quite interesting and relevant to understanding host-parasite co-evolution.

Fig 2.

Panel b images are WAY too small to see the structure clearly. This is a key result and should be much larger.

Panel C is quite confusing. I suggest coloring IgM chains in different shades of green or different colors and better labeling each chain. It is not clear which letters correspond to what. Also, the tailpieces do form beta sheets in the image and I think it would be best if they are depicted that way. Sometimes Pymol or other programs don't recognize sheets but one can specify that they are shown that way in a cartoon (if in fact you have evidence for hydrogen bonding characteristic of a beta-sheet). Under the image of the tailpieces, authors show an image signifying a S-S bond. However, it is not clear where on the structure this is located. As shown, it implies its part of the central tailpieces but in fact this S-S is distal from the tailpieces and is linking adjacent Fcs. If authors indeed want to point this out, they should make this a separate panel (d), label the domains and show where on the structure it is (e.g. by boxing a region in panel b, etc.).

Fig 3.

Panel a is too small. Also, the depiction of and labeling of different IgM chains is not clear at all. I suggest in all figures authors make each IgM chain a different color (perhaps a different shade of green) and then the sticks in the insets could also be colored according to the chain color (also it will then be clear which parts of VAR2CSA interact with which tailpiece if referring back to Fig. 2c). Additionally, the red and tan domains are not labeled. What are those? In all your figures authors

might simply labeling by including a color-coded key to the various complex components.

Fig. S3. It would be nice if authors included some images of representative fit of the model sidechains to the map; ideally this would be included for side chains of residues forming the interface in Fig. 3.

REVIEWER COMMENTS

Reviewer #1 (Remarks to the Author):

Please see attached marked-up manuscript.

We have responded to each comment in the the marked up manuscript. Please see that we have incorporated most of the suggestions.

In addition:

The absence of a "Track-Changes" version of the manuscript makes it quite laborious to check the changes between the original and the current manuscript.

Re. my original Major Point 4: The data provided is an improvement. However, assessment of VAR2CSA expression by flow cytometry would be much better (Supplementary Fig. 1a documents expression for just a very few IEs, leaving it completely unknown how many of the IEs in the culture actually expressed VAR2CSA.

We have added the flow cytometry data in the revised manuscript. Please see fig. S1c. Here, most of the infected erythrocytes are VAR2CSA positive.

Re. my original Major Point 5.1: The revised figure is an improvement.

We appreciate reviewer's comments.

Re. my original Major Point 5.2: There is still no information on how many IEs were added to the assay, only how many that stuck. To illustrate what I mean, let us assume that 100 IEs were added, and 10 bound. This would be 10% adhesion. However, if 10,000 IEs were added, the same 10 bound IEs would only represent 0.1%. Information on how many IEs were added to the assay is therefore critical to evaluate the results.

We added 2×10^6 IEs in the assay and mentioned in the methods section pg 15 line 468.

Further, we routinely select the parasites on CSA to maintain maximal expression of VAR2CSA on the surface of IEs that can easily be seen by FACS in our current revised version (Fig. S1c) where more than 90-95% of the parasites are VAR2CSA positive. Moreover, we are also aware that the reviewer has raised this concern as the reviewer is concerned that we are showing reduced binding to parasites grown in NHI. Therefore, in order to add equal parasites we take the CS2-albumax culture at early ring stage (4-5h post-invasion), wash well and and divided the culture in

CS2-albumax and CS2-NHI and then perform the binding assay. Thus in this way we ensure that we add equal number of parasites (2×10^6 IEs). We have also counted equal number of fields from each binding experiments to include averaging and avoid any bias.

Re. my original Major Point 5.4: The blot data in the right side of Fig. 1d need labeling indicating the location and identity of the relevant bands (which are hard to see and/or appear strangely dot-like).

We have denoted the complex band by indicating through arrow. The complex was obtained after gradient fixation in grafix and ran on NATIVE PAGE. The gel was transferred and blotted using anti-IgM (CH4) and anti-VAR2CSA antibodies. The band denoted by arrow was positive for both antibodies suggesting the presence of both the proteins in the complex.

Re. my original Major Point 6: I find it highly disturbing that the revised manuscript remains completely silent on the Ji et al. study, despite my explicit request to discuss their highly discordant -and as far as I can see, nevertheless quite solid – data.

We have included Ji *et al*, in our discussion to mark that interfacing residues are same but the discordance in the two structures in terms of stoichiometry in which IgM and VAR2CSA interact could be due to use of truncated IgM by Ji et al. Please see page 10 Line # 313-317 in the revised manuscript.

Re. my original Major Point 7: The study by Stevenson et al. Cell Microbiol 2015 studied the interaction between HB3VAR60 and IgM, not α 2-macroglobulin, as claimed by the authors in the response.

We apologize for the mistake. We have taken a note of it.

Re. my original Major Point 10: Still several instances, where the suggested clinical relevance of the proposed impairment adhesion of VAR2CSA-positive IEs due to IgM needs tempered (please refer to the attached marked-up manuscript).

We have made necessary changes as suggested by the reviewer. Almost all suggestion marked in the attached pdf by reviewer have been incorporated. Please also look at the attached response marked in the pdf that you marked for us.

New Major comment:

1. Please provide details regarding the “anti-VAR2CSA antibody” mentioned several times in the text

The protocol was described in the Methods section page 16, line 481-489.

Reviewer #2 (Remarks to the Author):

In this manuscript, authors Goel, Akhouri and Skoglund present data detailing a cryoEM structure of human IgM in complex with the Plasmodium falciparum protein complex VAR2CSA. VAR2CSA is known to contribute to placental malaria by promoting localization of P. falciparum-infected erythrocytes (IEs) to chondroitin sulfate proteoglycans. VAR2CSA is also known to bind to human immunoglobulin (Ig) M, although the functional consequences (for the host and parasite) of this interaction have remained unclear. The structure together with supporting biological experiments reveals that two VAR2CSA can bind IgM, details the IgM- VAR2CSA molecular interface and demonstrates that IgM binding to VAR2CSA limits its interaction with host factors, including its receptor and proteoglycans. Based on data, authors contend that IgM may function to “modulate the sequestration IEs on syncytiotrophoblast surfaces.” Overall, I think this work presents significant findings relevant to understanding IgM and VAR2CSA functions and more broadly, IgM interactions with parasite proteins and factors affecting the pathophysiology of P. falciparum infection and associated malaria disease. While the topic and data are interesting, structural analysis and presentation (e.g. figure content) of structural findings appear incomplete and need to be improved. As presented, it is difficult to evaluate if the data fully support the conclusions that authors have drawn. Additionally, I suspect that structural findings could include insights authors have not presented, and thus further attention to data analysis should not only improve, but perhaps expand upon, the conclusions presented and allow for deeper discussion. I anticipate that authors should be able to address these weaknesses in a revised manuscript; I have listed minor and major comments below.

General comments on the introduction:

Authors may want to change immunoglobulin “mu” to immunoglobulin “M”. I believe there are multiple, historic ways to describe IgM, but in practice I think most often “mu” typically refers just to the heavy chain whereas “M” would refer to the entire antibody, which seems to be what is implied in cases where this term is used.

We have changed ‘mu’ to M

Some basic introduction to IgM would be helpful (e.g. that it contains a JC and five IgM monomers, each with two heavy chains and two light chains). This will help the

audience later when anti-JC western blots are performed and when the structure is described (see specific comments below).

We have added information about IgM in the revised manuscript in result section before discussing about IgM in our structure page 5 line#144-150.

Major/minor Comments by line:

Line 59. Suggest changing “parasite” to P. falciparum and defining the PfEMP1 acronym.

We have changed according to reviewer’s suggestion in the revised manuscript page 2, line#64.

Line 73. Add a space between VAR2CSA and “is”

Based on the reviewer 1 suggestion, we have removed the sentence, therefore this mistake is not present in the current format.

Line 82. I am not sure if it matters if binding is occurring to immune or non-immune IgM? Both would be interesting and relevant.

Previous studies have shown the importance of non-immune IgM in mediating virulence of malaria parasites and have performed these assays only with non-immune IgM. The role of immune IgM has not be established in malaria parasite biology. However, we used non-immune IgM because of the established literature.

Line 83. Suggest removing the word “bulky” because it’s a relative description. It’s also not clear if the size of IgM would matter (nor have authors introduced the size or shape of IgM at this point in the text; see comments in results on this point) in the context of VAR2CSA binding to CSA.

We agree with the reviewer and have removed the term ‘bulky’ from the sentence. Regarding size and shape of the IgM: we have included more details in figure 4c to show how big is IgM with respect to VAR2CSA and describe about IgM in detail in page 5 line#144-150.

Lines 151-155. Authors may want to clarify that the high-resolution structure they are citing is Secretory (S) IgM, not serum IgM and thus, that published structure contains the secretory component (SC).

We have included citation for pdb 6KXS (reference 24).

Line 156. Authors state that VAR2CSA interacts with the core of IgM and they say they “focus” on residues 561-568; however they do not actually provide a description of the entire VAR2CSA-IgM interface. Here starting the paragraph by highlighting all IgM domains contacted by VAR2CSA and perhaps the total buried surface area would give the audience an overview before they start to focus on the tailpieces (which I don’t think form the entire interface, but the figures are so small I cannot tell). In sum, it is unclear if authors have “cherry picked” interactions to discuss or if fact they have mentioned all that are present in their structure (maybe VAR2CSA only contacts the tailpieces, maybe not).

We apologize for the confusion. In revised manuscript we mention that VAR2CSA interacts with the only C μ 4 and not with the tailpiece of the core. Although we did not cherry pick the interactions, but we agree with the reviewer’s comments. Therefore, we now used PDBePISA for analysis of interface and buried surface area. We have discussed the results of the findings in the text page 6-8 and Fig 3 and Supplementary Fig 6.

Line 158. Here the term tailpieces is used for the first time without definition (it should be stated that these are C-terminal extensions following the Ig-fold of C μ 4). Again, as mentioned above, it would help if IgM domain organization and structure were mentioned in the introduction.

We have included reviewer’s suggestion and therefore describe the overall organization of IgM in the results section (page 5 line#144-150) where we describe domains and the tailpiece for the readers. As suggested by reviewer, we have also stated that tailpiece are C-terminal extensions following the Ig fold (page 5, line 149-150). Additionally, we have also added reference for more information.

Line 167 (and 172). Authors state that VAR2CSA engages two “identical” surfaces on opposite faces of IgM. This may be true, but the statement is not justified without presenting a structural alignment and report of RMSD of interfacing residues on both sides. While the IgM tailpieces each share the same sequences, it is well established that the JC and its binding to IgM (or IgA) introduces asymmetry into the resulting complex so it is not clear the two sides are in fact “identical” (for example does VAR2CSA contact the JC?).

We appreciate the reviewer’s comment. Based on your suggestion, we calculated RMSD of interfacing residues of each side of VAR2CSA-IgM complex that is 0.837 Å. We have also added the data as Supplementary Fig. 5. Further we did not observe interaction of JC with VAR2CSA.

Line 176-177. Authors mention that loop regions in IgM and VAR2CSA mediate the interface and cite Fig 2, but images in Fig 2 are very small and one cannot clearly see this interface.

We apologize for this. We have added the figure in revised manuscript showing the loop area in the figure 2 (Fig. 2d).

Line 179-181. Authors cite Fig 3a and discuss membrane proximal region and N-terminal regions of VAR2CSA however the figure doesn't provide any visual reference to these regions so it is not clear what they are referring to.

We have addressed this issue in revised manuscript in fig 3a where DBL3x and DBL5ε domains are labeled. In the revised manuscript, we have also removed the text for clarity of the reading.

Line 183. Authors note that the conformation of IgM-VAR2CSA is different than the published apo and CSA-bound VAR2CSA structures. HOWEVER, figures or RMSDs demonstrating this are not reported. This is necessary because otherwise we have no proof that their structure is different and these conformation differences could be quite interesting and relevant to their conclusions. Structural analysis of the differences among these structures should be added at least as a supplemental figure but maybe even main text would help, especially because on lines 193 and 197 authors come back to discussion of differences between their structure and the apo structure and claim there is a conformational difference but there are NO figures or any presented analysis or quantification showing that they are different. To appropriately describe a conformational difference, analysis will likely have to comment on the interfaces between VAR2CSA components and differences among reported structures containing VAR2CSA.

We appreciate reviewer's comments. We performed more analysis of the structures and compared apo-VAR2CSA structure (7B52+7NNH) and VAR2CSA in our complex. We observed that there is a huge conformational change in the DBL5ε domain. We have added the data in the revised Fig 3a and discussed in the text page 6, line # 174-191.

Line 186 and 189. The terms "stiff" helix and "thin" linker are not biophysically appropriate. Authors have not measured the stiffness of the helix nor would one expect some linkers to be "thin" while others would be "thick"???

We agree with the reviewer. We have removed these terms.

Line 198-199 Authors state "Since we observed the formation of a new core accompanied by conformational changes;" a new core and conformational changes compared to what? I suspect authors are referring to the other structures they have cited (but not shown a comparison to); however the way this is written is not clear.

We have removed the sentences and presented and discussed the data with RMSD values in Fig. 3 and discussed in the text page 6, line # 174-191.

Lines 232-233. Authors state “It is highly likely that bound IgM on IEs would mask VAR2CSA owing to its bulky size and the formation of a new core structure in the IgM-VAR2CSA complex”. Authors have presented no justification for this statement. As presented, it’s a hypothesis and should be worded accordingly. As noted above, some discussion of the dimensions of IgM need to be introduced in the introduction or here. Authors could conduct modeling and specify potential geometry and dimensions based on modeling to support this hypothesis but that may be beyond the scope of this work.

Based on reviewer’s suggestion, we have revised figure 4c and included the overall dimension of IgM that includes unresolved part of IgM in our structure with respect to VAR2CSA (Fig 4c). These composite model suggests that IgM would extend much further than VAR2CSA and therefore will create a wall. In order to understand more about this extension, we refined the map at binning 2 and multibody refinement. We were not successful in getting better map by multibody refinement. However, the refinement at higher binning gave us a map which suggested the potential role of IgM in masking the CSA binding region in DBL2x domain of VAR2CSA (please see below). Since the resolution was low and we could not model this region even after including larger dataset or by multibody refinement, we resorted on functional assays to understand the role of IgM in masking of CSA binding site. Though our functional assays provides strength to this theory by reduced staining of VAR2CSA on IE surface or 50% loss of binding to its receptor. However, we agree to the reviewer that we could abstain from writing masking. Therefore, we have removed the word masking from the manuscript.

However, if reviewers think that this piece of information could enhance the quality of manuscript or benefit research community and needs to be included, we will be very happy to do that.

Fig. Orthogonal views of map of IgM-VAR2CSA complex refined at bin2. IgM-VAR2CSA complex fitted with VAR2CSA(pdb 7B52+7NNH) and IgM(pdb 6KXS without secretory domain). CSA binding pocket in DBL2x is labeled and C μ 2-C μ 1 region in IgM extend in the plane of VAR2CSA in alternate directions and possibly cover CSA binding pocket in VAR2CSA

Lines 236 -237. I think authors need to be careful about using the term “masking”. Their data do not really support any number of possible mechanisms for how IgM might block VAR2CSA binding to CS2.” Authors observe less staining of VAR2CSA on CS2 IEs in NHI-plasma compared with albumax; so indeed, something is limiting the interaction but whether it is a change in kinetics, the “large size” of IgM or steric occlusion of the binding site (independent of the size of IgM) on VAR2CSA has not been worked out. I am also curious if one or two VAR2CSA always bind to one IgM in vivo? How many copies of VAR2CSA might need to be bound by IgM? This circles back to my comment earlier comment wondering if the binding sites on both sides of IgM are indeed identical? Authors discuss that one or two VAR2CSA could bind later in the discussion (where I have added a few more comments on their masking hypothesis).

We have removed masking from the manuscript. The low resolution map of IgM-VAR2CSA suggest steric occlusion of the binding site (fig above). We have modified the text accordingly.

Regarding how many VAR2CSA are bound to IgM in vivo; based on single molecule fluorescence study, the spacing between 2 VAR2CSA is such that an IgM would be in contact with neighboring VAR2CSA on IE surface when bound to VAR2CSA. We have discussed it in the manuscript as well (page 10 line # 300-308). However, it is beyond the scope of this study to resolve this issue due to lack of available technique.

Regarding identical interaction of VAR2CSA on two sides of IgM, we found that they are identical residues. We have included the results of the interacting interface, the residues interacting with both VAR2CSA in Fig. 3 and Fig S6. We have also provided RMSD of IgM-VAR2CSA (chain I) with IgM-VAR2CSA (chain M) in the Supplementary Fig 5. Additionally, we have provided the overall picture of interacting residues on both sides of IgM for better perspective (Supplementary Fig. 6).

Line 274 The wording “VAR2CSA molecules bound to the opposite faces of IgM begin their interaction at approximately 90 apart” is confusing. The figure clarifies this some; however I would phrase this in terms of where the VAR2CSA are located as opposed to say “beginning,” which implies that authors are discussing the order or timing of a process. The structure is static and the sentence as written implies that the binding mechanism is beginning with VAR2CSA binding in one way and

continuing in another way (which I don't think is what the authors want to say here). I actually think some of the information on the structure presented in this paragraph (angles of binding and location of the termini, etc.) might be better suited for the results section, where I have indicated that authors should more completely describe the overall structure. If this approach is implemented, the other contents of this paragraph can remain in the discussion to describe the biological significance of the findings.

We indeed did not imply the kinetics of interaction of domains with IgM, therefore we have removed the word 'beginning' to avoid any confusion. We have moved the text of angle and distance in the result section as suggested by the reviewer. We have also removed 90° apart from the text. Here, in the revised manuscript we have measured angle between same residue on two VAR2CSA on the opposite face, while including a point in IgM core equidistant from both residue. Please see revised fig. 4c

Line 287. Authors state that IgM is larger than VAR2CSA and therefore it could mask VAR2CSA; however large size doesn't not imply masking nor do authors quantify this apparent size difference in any way. They need to do something to better describe this to the audience and describe how the masking would occur. Which domains of VAR2CSA would likely be masked? Would IgM Fabs contribute? Would IgM N-linked glycans contribute? Because the Fabs are not visible in the structure, most of their structure figures do not give the impression that IgM is larger than VAR2CSA- particularly Fig. 4c! Perhaps Authors can make a schematic illustration of the hypothesis to include in the figure?

We have removed the word masking. In order to be more clear, we changed to steric hinderance due to large size. Accordingly, we have provided a composite figure (fig. 4c) in the revised manuscript where we have shown the part of IgM that is not resolved in the structure but is still part of the complex. Thus in fig 4c we show that overall IgM is >35 nm and VAR2CSA is much smaller than that.

We have a low resolution map of IgM-VAR2CSA complex where we see Fabs oriented in alternate directions unlike the planar C_μ4 that could further cage VAR2CSA bound to it. Based on this, we speculate that Fabs may also contribute towards reduction in binding of VAR2CSA to its receptor as well as its reduced staining in our FACS assay. However, due to lack of atomic model we are choosing to abstain from commenting about it in the manuscript.

lines 304-306. Authors are missing a reference for the cryo-ET of PfEMP1 in complex with IgM cryoET and I cannot make further judgement of the contradiction they raise without seeing the data. I think this will be important to 1) highlight the contradiction between current model and previous study and 2) support the binding model proposed in Fig. 4d. Overall the discussion could better link their data and model and therefore

together with the issues noted above, it is a bit hard to be sure if their data do indeed support the model they propose.

We have added the reference in the revised manuscript.

Other comments:

Results or discussion:

Authors fail to discuss how their structure might differ from secretory (S) IgM structures, which contain the secretory component (SC) and specifically whether or not the VAR2CSA overlaps with the SC binding site. Although it's unlikely that SIgM and VAR2CSA would ever cross paths (SIgM is located in mucosa, not blood), it's possible VAR2CSA evolved to bind at sites overlapping with SC, which would be quite interesting and relevant to understanding host-parasite co-evolution.

We have compared the structure of secretory IgM with the IgM-VAR2CSA complex (fig below) and observed that SC has all β architecture which is opposed to majorly α -helical VAR2CSA. Also they do not align with each other in the way they interact with IgM. Although we have not included the figure in our manuscript but if the reviewer suggest to include as a supplemental figure, we would be happy to include it.

Fig 2.

Panel b images are WAY too small to see the structure clearly. This is a key result and should be much larger.

We have improved the quality of the figure.

Panel C is quite confusing. I suggest coloring IgM chains in different shades of green or different colors and better labeling each chain. It is not clear which letters correspond to what. Also, the tailpieces do form beta sheets in the image and I think it would be best if they are depicted that way. Sometimes Pymol or other programs don't recognize sheets but one can specify that they are shown that way in a cartoon (if in fact you have evidence for hydrogen bonding characteristic of a beta-sheet). Under the image of the tailpieces, authors show an image signifying a S-S bond. However, it is not clear where on the structure this is located. As shown, it implies its part of the central tailpieces but in fact this S-S is distal from the tailpieces and is linking adjacent Fcs. If authors indeed want to point this out, they should make this a separate panel (d), label the domains and show where on the structure it is (e.g. by boxing a region in panel b, etc.).

We have colored monomers of IgM in different shades figure 2 and 3 and Supplementary Fig. 4 in the revised manuscript. Also, in the revised manuscript, we have moved the tailpiece arrangement of IgM at the core that is colored in different shades to supplementary fig 4. Since Cys414 image was a repetition of published reports, based on reviewers suggestion, we removed the figure highlighting with S-S bond between Cys414 .

Fig 3.

Panel a is too small. Also, the depiction of and labeling of different IgM chains is not clear at all. I suggest in all figures authors make each IgM chain a different color (perhaps a different shade of green) and then the sticks in the insets could also be colored according to the chain color (also it will then be clear which parts of VAR2CSA interact with which tailpiece if referring back to Fig. 2c). Additionally, the red and tan domains are not labeled. What are those? In all your figures authors might simply labeling by including a color-coded key to the various complex components.

We appreciate reviewer's suggestion. We have modified the figure with the colored IgM chains (Fig. 2c, Supplementary Fig 4) and the interacting interface with VAR2CSA (Fig. 3b and supplementary fig 6).

Fig. S3. It would be nice if authors included some images of representative fit of the model sidechains to the map; ideally this would be included for side chains of residues forming the interface in Fig. 3.

We have added the representative fit of the model with side chains in supplementary fig. S9.

Reviewers' Comments:

Reviewer #1:

Remarks to the Author:

The revised manuscript is a major improvement. My main remaining concern is the authors' inference regarding the *in vivo* relevance of the study, which I find speculative.

What the authors have shown is detailed structural evidence that the Fc-dependent IgM binding to VAR2CSA involves the C μ 4 domain of IgM and the DBL3X and DBL5 ϵ domains of VAR2CSA. They furthermore present evidence that this interaction interferes with the ability of specific IgG to label VAR2CSA-positive IEs and with the adhesion of the IEs to CSA *in vitro*. These are important findings in their own right, and I think the repeated (but unsupported) suggestion that the IgM binding to VAR2CSA serves to impair placental IE adhesion is unnecessary. I therefore strongly suggest a further revision of the title to focus on what is actually shown in the study, possibly along the lines of "Molecular mapping of the Fc-dependent interaction of IgM with the Plasmodium falciparum adhesive protein VAR2CSA". I also suggest removing similar conjectural "*in vitro* to *in vivo*" extrapolation scattered in the text, e.g., L. 81-2 ("... indicating... interaction", L. 273-5 ("However,... with IgM") and L. 343-4.

There are important differences between *in vitro* findings and their potential *in vivo* relevance. While I concede that the authors did find that adhesion of IEs to CSA *in vitro* was impaired in the presence of IgM compared to its absence, the *in vivo* interaction between IEs and host receptors clearly always occur in the presence of ample amounts of IgM. They have not shown that IgM inhibits the sequestration of Plasmodium falciparum-infected erythrocytes (IEs) in the placenta, and when they claim that (particularly in the manuscript title), the question is in comparison to what relevant (*in vivo*) scenario?

The authors themselves are obviously aware of the above issue. Thus, the final sentence of the abstract is much more circumspect than the title and is much better in line with the evidence they present in their study. The reason I am harping on this is not least that the literature is replete with study titles claiming to establish something that then quickly becomes (often unjustified) gospel, particularly if published in an influential journal. That, in turn, can make it unnecessarily difficult to set the record straight in subsequent studies (because "it is already well known that...").

Re. my original Major Point 4: Suppl. Fig. 1c is a major improvement. However, a panel showing labeling with pre-bleed (or irrelevant control) rabbit IgG, purified from serum in the same way as their rabbit anti-VAR2CSA IgG would be an additional major improvement.

Reviewer #2:

Remarks to the Author:

For the most part, authors responded to prior comments in a point-by-point fashion and made changes to the manuscript and figures as suggested. However, I found it VERY difficult to review this without a "tracked changes" version of the manuscript (this was mentioned by a previous reviewer responding to the initial submission as well). Regardless, there are still issues that need to be addressed and perhaps a few places where comments were misinterpreted. In sum, while improved, aspects of the work are still not clearly communicated making it cumbersome to review the science.

New comments by line:

Line 143. Remove word "secretory". Also the term "(whole molecule) is confusing. According to construct information provided in Fig. 2a, authors used polymeric IgM lacking secretory component and therefore they have not used "secretory IgM." They do appear to have used pentameric IgM. Authors may have been trying to distinguish their soluble, pentameric IgM from an IgM BCR. If so this needs to be clarified; as written the terminology is not ideal and is confusing.

Line 157 1)

The information in the paragraph is helpful but would be better suited for the introduction or tied to their data. For example, authors could clarify here where VAR2CSA binds. Lines 170-172 attempt to do this but do not clearly state WHERE (Which motif) its binding to. Furthermore, the writing in this paragraph is a bit redundant and could be simplified.

Line 178. I believe this should be "Supplementary Fig. 5" instead of "Supplementary Fig. 5a".

Line 184. Remove "an".

Lines 184-186. How RMSD is calculated is not included in the Methods section. DBL1X and ID3 domains are not labeled in the Fig. 3a.

Lines 188-190. Here authors only discuss "drifts" of three residues and in the writing do not really state where these residues are located. Two are close to the interface and one away from interface but again the overall direction of "drift" is not well defined. More evidence and figures would be helpful. I suggest that authors include a supplementary figure (or an additional panel to supplementary Fig. 6) to show more residues that "drift" in the IgM-bound VAR2CSA compared to apo-VAR2CSA. Also, the term "drift" is not molecularly appropriate. Authors are simply comparing differences between two conformations and should use scientifically appropriate terminology to describe this.

Lines 194-197. Proper figures need to be cited.

Lines 205. Change "chain E, G and K" to "chain E, chain G, and chain K" or "chains E, G, and K".

Lines 206. Change "F, D and B chain" to "chain F, chain D, and chain B" or "chains F, D, and B".

Lines 207. Authors need to specify that these chains belong to IgM.

Lines 212-215. Authors need to specify which chain and complex these residues belong to.

Lines 220-222. Authors need to specify which chain and complex these residues belong to.

Lines 225-228. Authors need to specify which chain and complex these residues belong to.

Lines 248-249. This may be a good place to include the figure shown in the rebuttal which shows CSA-binding pocket blocked by CH1-CH2 domains (said figure is shown below). Authors should consider using this in Figure 4; perhaps replacing the current Fig. 4c right panel. Note that Figure 4c is also slightly inaccurate- the schematic just indicates where the Fabs are, not CH2 (this is why an alternative figure would be better). It's also nice that the CSA binding pocket is labeled in the rebuttal figure.

Line 294. It should be specified chain E refers to chain E of IgM.

Line 295. Figure 4c should be cited here.

Line 300. The number "15.7nm" is inconsistent with the label in Fig. 4c.

Line 304. It should be noted that IgG is monomeric.

Lines 305-306. It is unclear which "IgM planar/binding region" authors are referring to. Proper figure(s) needs to be cited and the said "dimension of 18.2nm" needs to be labeled.

Lines 333-334. Proper figure needs to be cited.

Other comments:

Page 5. The authors indeed added some basic information on IgM but the description is added after the first mention of JC and C μ 4. I suggest moving the introduction to before line 128.

Fig.2b. DBL6e domain is not labeled.

Fig. 2d. 1) There are some distances labeled in Fig. 2d that are too small to identify. Are these labels here on purpose or by mistake? 2) In panel Fig. 2d, the radius of side chains and backbones are very close. It is hard to distinguish one from another. Authors may consider enlarging these two frames and make the side chain representation thicker.

Fig.3a.

1) In the left panel, there are some indistinguishable labels on DBL5e. They should be either enlarged or removed.

2) DBL1X, ID3 domains are not labeled.

3) Label "DBL5e" is placed in between the DBL5e domain of the left panel and the DBL3x domain of the right panel; therefore, it is confusing as to which domain the label refers to.

4) Domain organization of IGM-bound VAR2CSA-2 is ambiguous in this panel. To change this, authors may want to consider applying the same coloring scheme from Fig. 2b to this panel.

5) DBL6e is not labeled but this figure is cited in lines 191-194 where DBL6e is discussed

Fig.4c. The right panel could be replaced by the figure that authors made in rebuttal (please see comments above).

Supplementary Fig. 5. How the RMSD was calculated is not described in Methods. For example, is this a backbone-only alignment? Which set of atom pairs are excluded from the 0.867Å calculation vs. the 1.258Å calculation).

Supplementary Figs. 7&8. I would suggest remaking these two figures using Esript.

REVIEWERS' COMMENTS

We thank both reviewers for their comments that led to improvement of the manuscript. Please find below the point-by-point response to comments.

Reviewer #1 (Remarks to the Author):

The revised manuscript is a major improvement. My main remaining concern is the authors' inference regarding the in vivo relevance of the study, which I find speculative.

What the authors have shown is detailed structural evidence that the Fc-dependent IgM binding to VAR2CSA involves the C₄ domain of IgM and the DBL3X and DBL5 ϵ domains of VAR2CSA. They furthermore present evidence that this interaction interferes with the ability of specific IgG to label VAR2CSA-positive IEs and with the adhesion of the IEs to CSA in vitro. These are important findings in their own right, and I think the repeated (but unsupported) suggestion that the IgM binding to VAR2CSA serves to impair placental IE adhesion is unnecessary. I therefore strongly suggest a further revision of the title to focus on what is actually shown in the study, possibly along the lines of "Molecular mapping of the Fc-dependent interaction of IgM with the Plasmodium falciparum adhesive protein VAR2CSA". I also suggest removing similar conjectural "in vitro to in vivo" extrapolation scattered in the text, e.g., L. 81-2 ("... indicating... interaction", L. 273-5 ("However,... with IgM") and L. 343-4.

There are important differences between in vitro findings and their potential in vivo relevance. While I concede that the authors did find that adhesion of IEs to CSA in vitro was impaired in the presence of IgM compared to in its absence, the in vivo interaction between IEs and host receptors clearly always occur in the presence of ample amounts of IgM. They have not shown that IgM inhibits the sequestration of Plasmodium falciparum-infected erythrocytes (IEs) in the placenta, and when they claim that (particularly in the manuscript title), the question is in comparison to what relevant (in vivo) scenario?

The authors themselves are obviously aware of the above issue. Thus, the final sentence of the abstract is much more circumspect than the title and is much better in line with the evidence they present in their study. The reason I am harping on this is not least that the literature is replete with study titles claiming to establish something that then quickly becomes (often unjustified) gospel, particularly if published in an influential journal. That, in turn, can make it unnecessarily difficult to set the record straight in subsequent studies (because "it is already well known that...").

Answer: We appreciate acknowledgement that manuscript has improved. Thank you so much for taking effort in helping us do that. We have removed the sentences you have suggested (please check the revised manuscript with

the track changes). However, regarding the title of the manuscript we are changing that to:

“Cryo-electron microscopy of IgM-VAR2CSA complex reveals IgM inhibits binding of *Plasmodium falciparum* to Chondroitin Sulfate A”

We strongly believe that it is not an extrapolation and there is no conjecture as we have removed placenta word from our title. We have done experiments with CSA and we are just sticking to that. Therefore, we should be able to mention it in our title (Please check Fig 4c). I hope this will be accepted without much problem.

Re. my original Major Point 4: Suppl. Fig. 1c is a major improvement. However, a panel showing labeling with pre-bleed (or irrelevant control) rabbit IgG, purified from serum in the same way as their rabbit anti-VAR2CSA IgG would be an additional major improvement.

Answer: We have repeated the experiment with IgG purified from pre-immune sera (Supplementary Fig. 1c).

Reviewer #2 (Remarks to the Author):

For the most part, authors responded to prior comments in a point-by-point fashion and made changes to the manuscript and figures as suggested. However, I found it VERY difficult to review this without a “tracked changes” version of the manuscript (this was mentioned by a previous reviewer responding to the initial submission as well). Regardless, there are still issues that need to be addressed and perhaps a few places where comments were misinterpreted. In sum, while improved, aspects of the work are still not clearly communicated making it cumbersome to review the science.

Answer: We apologize for the lack of track changes previously. In the revised manuscript we have track changes.

New comments by line:

Line 143. Remove word “secretory”. Also the term “(whole molecule) is confusing. According to construct information provided in Fig. 2a, authors used polymeric IgM lacking secretory component and therefore they have not used “secretory IgM.” They do appear to have used pentameric IgM. Authors may have been trying to distinguish their soluble, pentameric IgM from an IgM BCR.

If so this needs to be clarified; as written the terminology is not ideal and is confusing.

Answer: We have removed secretary, whole molecule from the manuscript. We have changed that to pentameric IgM. We apologize for our mistake (Line #78).

Line 157 1)

The information in the paragraph is helpful but would be better suited for the introduction or tied to their data. For example, authors could clarify here where VAR2CSA binds. Lines 170-172 attempt to do this but do not clearly state WHERE (Which motif) its binding to. Furthermore, the writing in this paragraph is a bit redundant and could be simplified.

Answer: We have modified line 157 (please check the revised manuscript) and also clarified to which region of IgM the VAR2CSA binds (line 154-168). Paragraph has been simplified.

Line 178. I believe this should be “Supplementary Fig. 5” instead of “Supplementary Fig. 5a”.

We apologize for our mistake, we have modified the text and referred Supplementary Fig.5. at line 182-183 .

Line 184. Remove “an”.

Answer: We have removed ”an” (now it is line #192).

Lines 184-186. How RMSD is calculated is not included in the Methods section. DBL1X and ID3 domains are not labeled in the Fig. 3a.

Answer: We have included details of RMSD calculation in our results section (Line 172-183) as well as methods section (Line 467-471). We have also included gradient of RMSD across molecule for clarity purposes (Supplementary Fig. 5).

DBL1X was a mistake and our sincere apology for that. It should be ID2-ID3 and we have labeled them in the comparison figure (Fig. 3a and Supplementary Fig. 6a and b).

Lines 188-190. Here authors only discuss “drifts” of three residues and in the writing do not really state where these residues are located. Two are close to the interface and one away from interface but again the overall direction of “drift” is not well defined. More evidence and figures would be helpful. I suggest that authors include a supplementary figure (or an additional panel to

supplementary Fig. 6) to show more residues that “drift” in the IgM-bound VAR2CSA compared to apo-VAR2CSA. Also, the term “drift” is not molecularly appropriate. Authors are simply comparing differences between two conformations and should use scientifically appropriate terminology to describe this.

Answer: We have mentioned location of these residues. drift word has been changed to shift. We have also included more residues (total 10) as per your suggestion in Fig.3a and Supplementary Fig.6b. We have added arrow to show the direction of shift and also added in the text (Please see line 196-203). Lines 194-197. Proper figures need to be cited.

Answer: We have cited Fig 2b in the revised manuscript (Line 208 and 211).

Lines 205. Change “chain E, G and K” to "chain E, chain G, and chain K" or “chains E, G, and K”.

Answer: we have modified the sentence as per your suggestion in the revised manuscript (line 223).

Lines 206. Change “F, D and B chain” to "chain F, chain D, and chain B" or “chains F, D, and B”.

Answer: we have modified the sentence as per your suggestion in the revised manuscript (line 227-228).

Lines 207. Authors need to specify that these chains belong to IgM.

Answer: we have modified the sentence as per your suggestion in the revised manuscript (line 231).

Lines 212-215. Authors need to specify which chain and complex these residues belong to.

Answer: We have modified as per your suggestion line 231-235.

Lines 220-222. Authors need to specify which chain and complex these residues belong to.

Answer: We have modified as per your suggestion line 241-245.

Lines 225-228. Authors need to specify which chain and complex these residues belong to.

Answer: We have modified as per your suggestion line 249-252.

Lines 248-249. This may be a good place to include the figure shown in the

rebuttal which shows CSA-binding pocket blocked by CH1-CH2 domains (said figure is shown below). Authors should consider using this in Figure 4; perhaps replacing the current Fig. 4c right panel. Note that Figure 4c is also slightly inaccurate- the schematic just indicates where the Fabs are, not CH2 (this is why an alternative figure would be better). It's also nice that the CSA binding pocket is labeled in the rebuttal figure.

Answer: We have removed Fig. 4c and added the rebuttal figure as Figure 4a to suit the flow as per your suggestion. We have also labeled CSA binding pocket.

Line 294. It should be specified chain E refers to chain E of IgM.

Answer: We have modified as per your your suggestion line line 318.

Line 295. Figure 4c should be cited here.

Answer: We have cited Fig. 4d (as previous Fig.4c is now 4d) line 321.

Line 300. The number "15.7nm" is inconsistent with the label in Fig. 4c.

Answer: We included the distance 15.7nm in the revised Fig. 4d. Please also check line 324.

Line 304. It should be noted that IgG is monomeric.

Answer: We have noted that IgM is monomeric and mentioned that in the line 329.

Lines 305-306. It is unclear which "IgM planar/binding region" authors are referring to. Proper figure(s) needs to be cited and the said "dimension of 18.2nm" needs to be labeled.

Answer: We have changed this to 'C μ 4 of IgM' to avoid confusion. We have mentioned distance in Fig 2c and have cited that (line 331). Fig. 4e is mentioned in the argument that if 2 VAR2CSA came from different parasite the distance would be much larger than 18.2nm (line 335).

Lines 333-334. Proper figure needs to be cited.

Answer: We have cited Fig. 4c line 359.

Other comments:

Page 5. The authors indeed added some basic information on IgM but the description is added after the first mention of JC and C μ 4. I suggest moving the introduction to before line 128.

Answer: We have moved this part in introduction line 77-83.

Fig.2b. DBL6e domain is not labeled.

Answer: We have modeled residues between ID2-DBL5e. Therefore no labeling for DBL6e in our atomic model.

Fig. 2d. 1) There are some distances labeled in Fig. 2d that are too small to identify. Are these labels here on purpose or by mistake? 2) In panel Fig. 2d, the radius of side chains and backbones are very close. It is hard to distinguish one from another. Authors may consider enlarging these two frames and make the side chain representation thicker.

Answer: We have removed the small labels. We have enlarged the Fig. 2d and side chains appear thicker in the modified Fig 2d.

Fig.3a.

1) In the left panel, there are some indistinguishable labels on DBL5e. They should be either enlarged or removed.

Answer: We have removed small labels.

2) DBL1X, ID3 domains are not labeled.

Answer: We have labeled DBL1X and other domains are colored in accordance with Fig.2b. and colors are mentioned in the figure legend.

3) Label "DBL5e" is placed in between the DBL5e domain of the left panel and the DBL3x domain of the right panel; therefore, it is confusing as to which domain the label refers to.

Answer: We have changed the figure and this problem is removed.

4) Domain organization of IGM-bound VAR2CSA-2 is ambiguous in this panel. To change this, authors may want to consider applying the same coloring scheme from Fig. 2b to this panel.

Answer: We have applied same color scheme as Fig.2b.

5) DBL6e is not labeled but this figure is cited in lines 191-194 where DBL6e is discussed

Answer: We have labeled DBL6e in revised Fig 3a.

Fig.4c. The right panel could be replaced by the figure that authors made in rebuttal (please see comments above).

Answer: We have removed Fig. 4c and added the rebuttal figure as Fig. 4a to suit the flow as per your suggestion.

Supplementary Fig. 5. How the RMSD was calculated is not described in Methods. For example, is this a backbone-only alignment? Which set of atom pairs are excluded from the 0.867Å calculation vs. the 1.258Å calculation).

Answer: We have included details of RMSD calculation in our results section (Line 174-183) as well as methods section (Line 467-471). We have also included gradient of RMSD across molecule for clarity purposes (Supplementary Fig. 5). We have also mentioned that it is only for c-alpha backbone.

In IgM-VAR2CSA monomer comparison, calculation for 2696 pruned residues and for all 3261 residues are mentioned in the results.

For Apo-VAR2CSA Vs IgM-VAR2CSA monomer the comparison is for pruned 580 residues and for further for 865 residues are reported in results. It is to be noted that in this comparison DBL5e is not considered by matchmaker due to large conformational shift which is beyond 10Å (Supplementary Fig.6a).

Supplementary Figs. 7&8. I would suggest remaking these two figures using Esript.

Answer: We have prepared a revised Fig. 8 and 9 (new figure number) using ESript.